# HIF-2α is indispensable for regulatory T cell function

Tzu-Sheng Hsu[1], Yen-Lin Lin[1], Yu-An Wang[1], Shu-Ting Mo[1], Po-Yu Chi[2], Alan Chuan-Ying Lai[2], Hsuan-Yin Pan[1], Ya-Jen Chang[2] & Ming-Zong Lai [1✉]

Hypoxia-inducible factor 1α (HIF-1α) and HIF-2α are master transcription factors that regulate cellular responses to hypoxia, but the exact function in regulatory T (Treg) cells is controversial. Here, we show that Treg cell development is normal in mice with Foxp3-specific knockout (KO) of HIF-1α or HIF-2α. However, HIF-2α-KO (but not HIF-1α-KO) Treg cells are functionally defective in suppressing effector T cell-induced colitis and inhibiting airway hypersensitivity. HIF-2α-KO Treg cells have enhanced reprogramming into IL-17-secreting cells. We show crosstalk between HIF-2α and HIF-1α, and that HIF-2α represses HIF-1α expression. HIF-1α is upregulated in HIF-2α-KO Treg cells and further deletion of HIF-1α restores the inhibitory function of HIF-2α-KO Treg cells. Mice with Foxp3-conditional KO of HIF-2α are resistant to growth of MC38 colon adenocarcinoma and metastases of B16F10 melanoma. Together, these results indicate that targeting HIF-2α to destabilize Treg cells might be an approach for regulating the functional activity of Treg cells.

---

[1] Institute of Molecular Biology, Academia Sinica, 11529 Taipei, Taiwan. [2] Institute of Biomedical Sciences, Academia Sinica, 11529 Taipei, Taiwan. ✉email: mblai@gate.sinica.edu.tw

CD4+CD25+ regulatory T (Treg) cells inhibit T cell activation and are essential for maintaining immune tolerance. Treg cells are classified into thymus-derived CD4+CD25+Foxp3+ regulatory T (tTreg) cells, peripheral regulatory T (pTreg) cells and induced regulatory T (iTreg) cells[1]. Treg cells are characterized by their expression of the master transcription factor forkhead box P3 (Foxp3), which dictates their development and function[2–4]. Persistent expression of Foxp3 is required throughout the lifetime of Treg cells to maintain their suppressive activities[5,6]. Significant efforts have been dedicated to establishing how Foxp3 gene expression is regulated[4,7]. Notably, the extent of methylation in the Treg-specific demethylation region (TSDR) of the Foxp3 gene determines Foxp3 stability[8,9]. Direct dendritic cell stimulation or IL-2 treatment promotes Treg stability through TSDR demethylation of Foxp3 (refs. [10,11]). Foxp3 protein stability is further regulated by several post-translational modifications, including ubiquitination, acetylation and phosphorylation[3,12–14]. Some Treg cells are converted into inflammatory effector cells following Foxp3 destabilization, which may lead to loss of their suppressive activity and expression of inflammatory cytokines such as IFN-γ or IL-17 (refs. [15–17]). There are reports that Treg cells can be reprogrammed into inflammatory cells in certain autoimmune diseases[18–22]. Recent studies have demonstrated that ex vivo expansion of Treg cells can be used to treat autoimmune diseases and transplantation rejection[23–25]. Maintenance of stable suppressive activity is essential for successful therapeutic applications of Treg cells. However, Treg cells contribute to immunosuppressive environments and are a major element inhibiting intratumor anticancer immunity[26–28]. Treg-selective deletion of essential Treg factors such as Neuropilin-1 (Nrp-1), suppressor of cytokine signaling 1 (SOCS1), enhancer of zeste homolog 2 (EZH2), Eos (IKzf4), or CARMA1 leads to Foxp3 downregulation and inflammatory cytokine secretion, conferring resistance to tumor growth in the host[20–22,29–33].

Hypoxia-inducible factor 1α (HIF-1α) and HIF-2α are master transcription factors that regulate physiological hypoxic responses[34–37]. In the presence of O2, HIF-1α and HIF-2α are hydroxylated at key proline residues by prolyl hydroxylase domain protein 2 (PHD2)/PHD3, leading to recognition by the von Hippel-Lindau (VHL)-containing E3 complex that ubiquitinates HIF-1α and HIF-2α for their proteasomal degradation. Inactivation of PHD2/PHD3 under hypoxic conditions stabilizes HIF-1α and HIF-2α proteins[38]. HIF-1α and HIF-2α are then heterodimerized with HIF-1β subunits to activate expression of target genes involved in hypoxic responses[34–37]. In addition, HIF-1α can be upregulated in T cells by continuous T cell receptor (TCR) stimulation under normoxic conditions[39,40].

The role of HIF-1α in Treg cells has been explored in many studies. HIF-1α has been shown to bind Foxp3 and to promote Foxp3 degradation, thereby inhibiting Treg cell differentiation[41]. HIF-1α regulates T cell metabolism and participates in glycolysis, thereby suppressing Treg cell development[42]. Foxp3 protein stability is increased by an insertional mutation that blocks binding of HIF-1α[43]. Consistent with those findings, VHL-knockout (KO) Treg cells lose their suppressive function and produce excess IFN-γ, whereas additional HIF-1α-KO restores Treg cells activity[18]. Similarly, persistent expression of HIF-1α by deleting PHD1, PHD2, and PHD3 in T cells leads to a significant increase in the ratio of IFN-γ+ effector T cells to Treg cells[30]. Previously, we also report that Treg cells become highly unstable in vivo in the absence of the E3 ligase deltex1 that downregulates HIF-1α[44], supporting the inhibitory role of HIF-1α on Treg cells. In addition, iTreg differentiation is inhibited by hypoxia, which can be reversed by HIF-1α deficiency[45], further confirming the suppressive activity of HIF-1α in Treg cell differentiation. By contrast, a putative hypoxia-responsive element is found on the promoter of Foxp3, and HIF-1α-KO CD4+CD25+ Treg cells fail to protect mice from effector T-cell-induced colitis[46]. Thus, the exact role of HIF-1α in Treg cell differentiation and action remains uncertain.

Most studies of the function of HIF-2α in immune responses have used myeloid cells. HIF-2α is induced by T helper 2 (Th2) cytokines during M2 macrophage polarization and specifically regulates expression of arginase-1 (ref. [47]). Lack of HIF-2α in the myeloid lineage results in decreased tumor-associated macrophage infiltration and alleviates tumor progression[48]. Loss of HIF-2α from myeloid cells also increases neutrophil apoptosis and reduces neutrophilic inflammation[49]. However, the function of HIF-2α in the differentiation and function of T cells and Treg cells is unclear[50].

Here, we show an unexpected role for HIF-2α in Treg cells. Development and the phenotype of HIF-2α-KO tTreg cells is normal, with unchanged in vitro suppressive activity. However, HIF-2α-KO Treg cells have an impaired ability to inhibit effector T cell-induced colitis and airway allergic inflammation. These defects in HIF-2α-KO Treg cells can be partly attributed to elevated HIF-1α expression, and further deletion of HIF-1α restores the suppressive function of HIF-2α-KO Treg cells. Consequently, mice with Foxp3-conditional knockout of HIF-2α are resistant to tumor growth. Our findings demonstrate an unanticipated requirement for HIF-2α in Treg function and suggest a potential approach to regulating Treg activity by targeting HIF-2α in Treg cells.

## Results

**HIF-2α-KO Treg cells have normal development and phenotype.** To study the role of HIF-2α in T cells, we generated mice with T cell-specific deletion of HIF-2α (Cd4CreHif2af/f) and Treg-specific knockout of HIF-2α (Foxp3CreHif2af/f). Deletion of HIF-2α was confirmed by RT-PCR (Supplementary Fig. 1). Thymus and peripheral T cell development was normal in Cd4CreHif2af/f mice (Supplementary Fig. 2a, b). Fractions of naïve and memory T cells were not affected by HIF-2α deficiency (Supplementary Fig. 2c). T cell proliferation, IL-2 production, and IFN-γ generation stimulated through CD3/CD28 were normal in Cd4CreHif2af/f T cells isolated from lymph nodes and spleen (Supplementary Fig. 3). Therefore, HIF-2α deficiency in T cells does not affect T cell development or T cell activation. We observed a similar outcome for T cell development in Foxp3CreHif2af/f mice (Supplementary Fig. 4). No splenomegaly or lymphadenopathy was observed in any of the Cd4CreHif2af/f or Foxp3CreHif2af/f mice.

For comparison with Treg cell development in Foxp3CreHif2af/f mice, we also assessed mice with Treg-intrinsic deletion of HIF-1α (Foxp3CreHif1af/f). The proportions of CD4+Foxp3+ T cells from lymph nodes, thymus and spleen were comparable between wild-type (WT, Foxp3Cre), Foxp3CreHif1af/f and Foxp3CreHif2af/f mice (Fig. 1a). We further confirmed normal development of Foxp3CreHif2af/f tTreg cells based on proportional expression of Helios[51] and Nrp-1 (Supplementary Fig. 4d). Treg cells phenotypes—characterized by the expression of cytotoxic T-lymphocyte-associated protein 4 (CTLA-4), folate receptor 4 (FR4), glucocorticoid-induced TNFR-related protein (GITR) and lymphocyte activation gene-3 (LAG-3)—were also similar between tTreg cells from Foxp3Cre, Foxp3CreHif1af/f and Foxp3CreHif2af/f mice (Fig. 1b). Upon differentiating the iTreg cells induced by different doses of TGF-β under normoxic conditions, we found no difference in the fractions of CD4+Foxp3+ T cells generated from Hif2af/f, Cd4CreHif1af/f or Cd4CreHif2af/f naïve CD4+ T cells (Fig. 1c). Expression of CTLA-4, FR4, GITR, and LAG-3 in these iTreg cells was also comparable (Supplementary

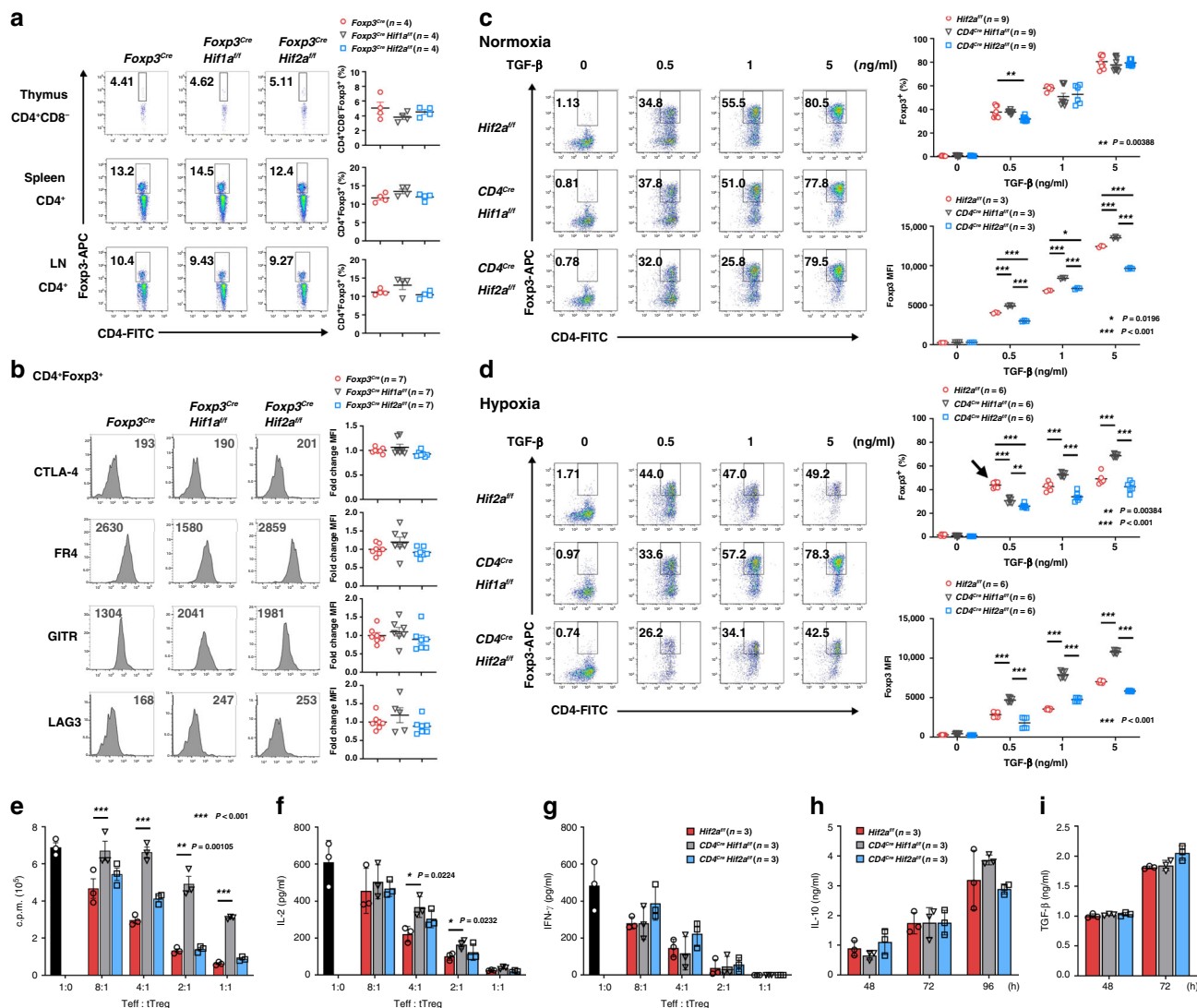

**Fig. 1 HIF-2α KO does not affect Treg cell development or in vitro tTreg suppressive activity. a** CD4⁺Foxp3⁺ T cell populations in thymus, spleen, and peripheral lymph nodes (LN) of *Foxp3^Cre* (WT), *Foxp3^Cre Hif1a^f/f*, and *Foxp3^Cre Hif2a^f/f* mice. Left, representative plots. Right, data expressed as mean ± SEM, $n = 4$ biologically independent mice. Red circle, *Foxp3^Cre*; black inversed triangle, *Foxp3^Cre Hif1a^f/f*; blue square, *Foxp3^Cre Hif2a^f/f*. **b** Expression of CTLA-4, FR4, GITR, and LAG-3 in *Foxp3^Cre*, *Foxp3^Cre Hif1a^f/f*, and *Foxp3^Cre Hif2a^f/f* tTreg cells. Left, representative traces of MFI. Right, ratios of the MFIs of HIF-1α-KO or HIF-2α-KO tTreg to WT tTreg cells are expressed as the mean ± SEM, $n = 7$ mice. **c**, **d** Naïve T cells from *Hif2a^f/f* (WT), *Cd4^Cre Hif1a^f/f*, and *Cd4^Cre Hif2a^f/f* mice were activated with anti-CD3/CD28 (2/1 μg ml⁻¹) with different doses of TGF-β and IL-2 (20 ng ml⁻¹) under normoxia **c** or hypoxia (1% O₂) **d** for 3 days. Left, representative plots of Foxp3 staining. Right, percentages of Foxp3⁺ cells and MFIs of Foxp3 are presented as the mean ± SEM, $n = 9$ (Foxp3⁺, **c**), $n = 3$ (Foxp3 MFI, **c**), $n = 6$ **d**. Symbols are the same as in **a** except red circle represents *Hif2a^f/f*. \*$P < 0.05$, \*\*$P < 0.01$, \*\*\*$P < 0.001$, as determined by two-way ANOVA. Arrow in **d** indicates a specific increase in Foxp3⁺ cells among WT iTreg cells. **e–g** WT CD4⁺CD25⁻ T cells (1 × 10⁵) were stimulated alone with anti-CD3 (1 μg ml⁻¹) and T cell-depleted splenic cells or together with *Hif2a^f/f* (WT), *Cd4^Cre Hif1a^f/f* or *Cd4^Cre Hif2a^f/f* tTreg cells at the indicated ratios. CD4⁺CD25⁻ T cell proliferation was measured by thymidine incorporation **e**, and the production of IL-2 **f** and IFN-γ **g** was determined 40 h later. Values are mean ± SD, $n = 3$ (**e–g**). \*$P < 0.05$, \*\*$P < 0.01$, by two-tailed unpaired *t*-test (**e**, **f**). **h**, **i** *Hif2a^f/f*, *Cd4^Cre Hif1a^f/f* and *Cd4^Cre Hif2a^f/f* tTreg cells were stimulated with anti-CD3/CD28 (4/2 μg ml⁻¹) and IL-2 for 48–96 h. IL-10 (**h**) and TGF-β (**i**) production were determined. $n = 4$ (**h**), $n = 4$ (*Hif2a^f/f*, *Cd4^Cre Hif2a^f/f*), 2 (*Cd4^Cre Hif1a^f/f*) **i**. Source data are provided as a Source data file.

Fig. 5a). However, we did observe a difference between HIF-1α-KO and WT iTreg cells differentiated under hypoxia. As reported previously[41,42], we found that hypoxia inhibited differentiation of WT (*Hif2a^f/f*) iTreg cells, which was reversed by HIF-1α knockout (Fig. 1d, *Cd4^Cre Hif1a^f/f*). Similar to *Hif2a^f/f* iTreg cells, *Cd4^Cre Hif2a^f/f* iTreg differentiation was suppressed by hypoxia (Fig. 1d). The inhibitory effect of hypoxia on iTreg differentiation was also observed for *Foxp3^Cre* and *Foxp3^Cre Hif2a^f/f* iTreg cells (Supplementary Fig. 6), even though the impact was not as profound as recorded for *Hif2a^f/f* and *Cd4^Cre Hif2a^f/f* iTreg cells

(Fig. 1d). These results suggest that HIF-1α inhibits iTreg cells at the stages of both Foxp3 expression and post-Foxp3 induction. Notably, there was an increase of WT (*Hif2a^f/f*) iTreg cells differentiated at a low concentration (0.5 ng ml⁻¹) of TGF-β under hypoxic conditions (Fig. 1d, indicated by arrow). This finding is similar to a previous study reporting that hypoxia increases iTreg differentiation under similar conditions (0.75 ng ml⁻¹ TGF-β)[46]. Overall, the development of HIF-2α-KO tTreg and iTreg cells is comparable with that of WT tTreg and iTreg cells, supporting that deficiency of HIF-2α does not interfere with

Treg cell development. Moreover, our experiments reveal that HIF-1α may fine-tune the differentiation of iTreg cells in a more complex way than previously recognized.

**HIF-2α-KO Treg cells have normal suppressive function in vitro**. Next, we determined the in vitro suppressive activity of tTreg cells. *Cd4^CreHif2a^f/f* tTreg cells proved as effective as WT (*Hif2a^f/f*) tTreg cells in inhibiting proliferation, as well as IL-2 and IFN-γ production, of effector T cells. By contrast, HIF-1α-KO tTreg cells were ineffective in inhibiting effector T cell proliferation and secretion of IL-2 (Fig. 1e–g). Production of IL-10 and TGF-β by *Cd4^CreHif2a^f/f* tTreg cells was indistinguishable from that by *Cd4^CreHif1a^f/f* and *Hif2a^f/f* tTreg cells (Fig. 1h, i). *Cd4^CreHif2a^f/f* iTreg cells exhibited levels of in vitro inhibitory activity comparable to WT iTreg cells (Supplementary Fig. 5b). Therefore, HIF-2α-KO tTreg cells and HIF-2α-KO iTreg cells exhibit normal in vitro suppressive activity.

**HIF-2α-KO tTreg cells have impaired ability to suppress colitis**. Treg cells that are defective in inhibiting effector T cells in vivo may display normal in vitro suppressive activity[44,52]. We examined the in vivo suppressive activity of WT, HIF-1α- and HIF-2α-KO tTreg cells. In this assay, adoptive transfer of CD45.1+ WT CD4+CD25− effector T cells into CD45.1+ RAG-1-KO mice led to colitis, and co-transfer of CD45.2+ Treg cells that are congenically different from effector T cells was used to evaluate the suppressive activity of Treg cells. Effector T cells-induced colitis was examined by body weight loss, diarrhea and shortened colon length (Fig. 2a–c). Histological examination of hematoxylin and eosin (H&E)-stained colon sections also revealed inflammatory cell accumulation and tissue damage in RAG-1-KO mice that had received CD45.1+ effector T cells (Fig. 2d). Co-transfer of CD45.2+ *Hif2a^f/f* (WT) tTreg cells protected RAG-1-KO mice from CD45.1+ effector T cell-induced body weight loss, colitis, diarrhea, tissue damage and inflammatory infiltration (Fig. 2a–d). HIF-1α-KO tTreg cells (*Cd4^CreHif1a^f/f*) were as effective as *Hif2a^f/f* tTreg cells in suppressing colitis and associated pathologies in RAG-1-KO mice receiving effector T cells (Fig. 2a–d).

Unexpectedly, a large number of *Cd4^CreHif2a^f/f* tTreg cells were unable to inhibit effector T cell-triggered colitis (Fig. 2a–d). We re-isolated the adoptive-transferred tTreg cells (CD45.2+) from RAG-1-KO mice after colitis induction and determined the expression of IL-17 and IFN-γ. We found that the fraction of IL-17+ or IFN-γ+ cells among the transferred HIF-2α-KO tTreg cells (CD45.2+) had increased relative to WT tTreg cells (Fig. 2e). In addition, recovery of the effector T cells (CD45.1+) transferred into RAG-1-KO mice revealed increased IL-17 production (Fig. 2f). A large proportion of RAG-1-KO mice that received *Cd4^CreHif2a^f/f* tTreg cells lost body weight and developed colitis, indicating that *Cd4^CreHif2a^f/f* tTreg cells have impaired in vivo suppressive activity.

The attenuated inhibitory activity of HIF-2α-KO tTreg cells was not due to decreased Foxp3 expression as proportions of Foxp3+ cells, determined by flow cytometry, were comparable between *Hif2a^f/f*, *Cd4^CreHif1a^f/f* and *Cd4^CreHif2a^f/f* tTreg cells upon recovery from RAG-1-KO mice (Fig. 2g). Therefore, HIF-2α deficiency, but not HIF-1α deficiency, impairs the in vivo suppressive function of tTreg cells that inhibit colitis.

**Reprogramming of HIF-2α-KO Treg cells into IL-17-secreting cells**. *Cd4^CreHif2a^f/f* tTreg cells recovered from RAG-1-KO mice with colitis displayed enhanced expression of IL-17 and IFN-γ (Fig. 2e), suggesting a susceptibility to being converted into inflammatory cytokine-producing cells. Therefore, we examined re-programming of *Cd4^CreHif2a^f/f* tTreg cells, stimulated by TCR/

CD28 in the presence of Th0-, Th1- or Th17-priming cytokines for 5 days. As previously reported[17,19], IFN-γ and IL-17 were produced in *Hif2a^f/f* tTreg cells upon such destabilizing treatments in vitro (Fig. 3a, b). IFN-γ expression was comparable between *Hif2a^f/f* and *Cd4^CreHif2a^f/f* tTreg cells upon Th0-, Th1- or Th17-priming (Fig. 3a). Foxp3 was downregulated to similar degrees in *Hif2a^f/f* and *Cd4^CreHif2a^f/f* tTreg cells after such treatments (Fig. 3c). By contrast, secretion of IL-17 by *Cd4^CreHif2a^f/f* tTreg cells was significantly higher than for *Hif2a^f/f* tTreg cells after stimulation under Th0, Th1 or Th17 priming conditions (Fig. 3b). Biased IL-17 generation (relative to *Hif2a^f/f* iTreg cells), but comparable IFN-γ-production, was also found among activated *Cd4^CreHif2a^f/f* iTreg cells destabilized with Th0-, Th1- or Th17-priming cytokines (Fig. 3d, e). We examined the expression of RORγt in activated *Cd4^CreHif2a^f/f* tTreg cells, but did not observe any increase in RORγt levels (Supplementary Fig. 7).

We also investigated if HIF-2α deficiency affects CD4+ T cell differentiation. Naïve CD4+CD25−CD44^lowCD62L^high T cells were activated with plate-bound anti-CD3/CD28 under Th1-, Th2-, and Th17-polarizing conditions, and then we determined the expression of IFN-γ, IL-4, and IL-17 in each lineage. There was no difference in Th1, Th2 or Th17 differentiation between *Hif2a^f/f* and *Cd4^CreHif2a^f/f* naïve CD4+ T cells (Fig. 3f). Therefore, HIF-2α is required to prevent Treg cells from being converted into IL-17-secreting cells, but it does not directly regulate Th1, Th2 or Th17 differentiation.

**HIF-2α-KO iTreg cells do not inhibit airway hypersensitivity**. We further extended our study to the in vivo inhibitory activity of HIF-2α-KO iTreg cells. In an allergen-induced airway inflammation assay (Fig. 4a), ovalbumin-induced airway resistance was prevented by administration of WT (*Hif2a^f/f*) iTreg cells, but not by *Cd4^CreHif2a^f/f* iTreg cells (Fig. 4b). Ovalbumn also induced infiltration of leukocytes into bronchoalveolar lavage fluid (BALF) (Fig. 4c). *Hif2a^f/f* iTreg cells decreased the amount of total leukocytes and eosinophils in BALF from ovalbumin-treated mice, yet *Cd4^CreHif2a^f/f* iTreg cells failed to exhibit any inhibitory effect (Fig. 4c). Ovalbumin-triggered IL-4 and IFN-γ in BALF was completely suppressed by *Hif2a^f/f* iTreg cells, in contrast to the ineffectiveness of *Cd4^CreHif2a^f/f* iTreg cells in the same assay (Fig. 4d, e). Levels of IL-6 were also fully suppressed by *Hif2a^f/f* iTreg cells, but were only weakly inhibited by *Cd4^CreHif2a^f/f* iTreg cells (Fig. 4d). Suppression of IL-5 and IL-13 in BALF was comparable between *Cd4^CreHif2a^f/f* and *Hif2a^f/f* iTreg cells. The differential in vivo suppressive activity of WT and HIF-2α-KO iTreg cells was also shown by histology. Ovalbumin re-challenge resulted in extensive inflammatory cell infiltration in perivascular and peribronchial regions, as well as bronchial wall thickening of the airway in mice sensitized with ovalbumin (Fig. 4f, OVA). Administration of *Hif2a^f/f* iTreg cells reduced inflammatory infiltration in lung tissue and bronchial wall thickening (Fig. 4f, OVA + *Hif2a^f/f* iTreg). HIF-2α-KO iTreg cells were less effective than OVA + WT iTreg cells in suppressing ovalbumin re-challenge-triggered allergic responses (Fig. 4f, OVA + *Cd4^CreHif2a^f/f* iTreg). Recovery of the transferred iTreg cells (CD45.2+) from allergic mice (CD45.1+) revealed that Foxp3 expression, measured by flow cytometry, was also comparable between HIF-2α-KO and WT iTreg cells (Fig. 4g). These results indicate that HIF-2α-KO iTreg cells exhibit defective in vivo suppressive activity.

**HIF-1α expression is increased in HIF-2α-KO Treg cells**. We examined the possible mechanisms by which HIF-2α deficiency impaired the suppressive function of Treg cells. HIF-1α expression

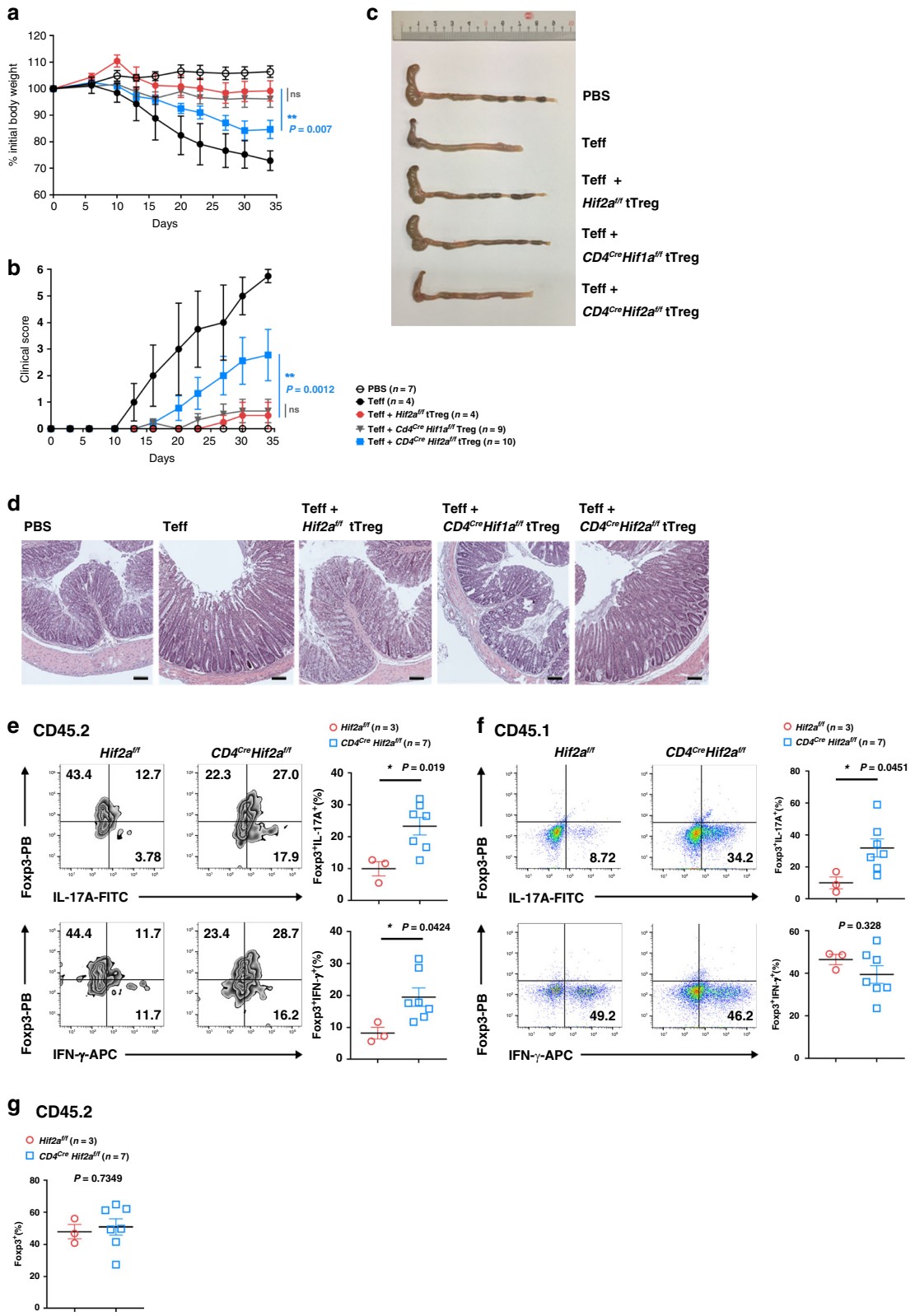

was increased in CD45.2⁺ $Cd4^{Cre}Hif2a^{f/f}$ tTreg cells recovered from CD45.1⁺ RAG-1-KO mice, relative to $Hif2a^{f/f}$ tTreg cells (Fig. 5a). Even though HIF-1α protein was not detectable in naïve T cells, increased $Hif1a$ expression was found in freshly isolated $Cd4^{Cre}Hif2a^{f/f}$ T cells and tTreg cells (Fig. 5b, c). HIF-1α protein can be induced under normoxic conditions when T cells are continuously activated[39,53], and we observed a more profound increase in HIF-1α protein in persistently activated $Cd4^{Cre}Hif2a^{f/f}$ CD4⁺ T cells compared to the corresponding $Hif2a^{f/f}$ CD4⁺ T cells (Fig. 5d). Activation by CD3/CD28 under hypoxia induced higher HIF-1α protein expression in HIF-2α-KO tTreg cells than the control tTreg cells (Fig. 5e). Activation of iTreg cells also led to greatly elevated HIF-1α levels in $Cd4^{Cre}Hif2a^{f/f}$ iTreg cells (Fig. 5f). In addition, inhibition of translation by cycloheximide illustrated

**Fig. 2 HIF-2α-KO tTreg cells are defective in suppressing colitis. a–d** CD45.1$^+$ CD4$^+$CD25$^-$ effector T (Teff) cells (4 × 10$^5$) were administered into sex-matched CD45.1$^+$ RAG-1-KO mice with or without CD45.2$^+$ tTreg cells (1 × 10$^5$) originating from *Hif2a$^{f/f}$*, *Cd4$^{Cre}$Hif1a$^{f/f}$*, or *Cd4$^{Cre}$Hif2a$^{f/f}$* mice. Body weight **a** and colitis scores **b** were assessed. Four out of the 10 RAG-1-KO mice that received *Cd4$^{Cre}$Hif1a$^{f/f}$* tTreg cells did not lose weight, but their data have been included in **a**, **b**. Open circle, PBS; black solid circle, Teff; red solid circle, Teff + *Hif2a$^{f/f}$* Treg; grey inverted square, Teff + *Cd4$^{Cre}$Hif1a$^{f/f}$* Treg, blue square, Teff + *Cd4$^{Cre}$Hif2a$^{f/f}$* Treg. Data represent a combination of two independent experiments, and were confirmed by another two independent experiments, each of which individually presented a pattern similar to that of the combined dataset. **$P = 0.007$ (**a**), 0.0012 (**b**), by repeat measures two-way ANOVA with Tukey's multiple comparisons test, with the Greenhouse-Geisser correction. ns, not significant. Five weeks after T-cell transfer, mice were sacrificed and colons were isolated **c**, and stained with hematoxylin and eosin (H&E) **d**. Micrographs are representative of the mice in each group. Bar indicates 100 μm. Results were reproduced in four independent experiments. **e–g** CD45.2$^+$ *Hif2a$^{f/f}$* or *Cd4$^{Cre}$Hif2a$^{f/f}$* tTreg cells were administered into CD45.1$^+$ RAG-1-KO mice together with CD45.1$^+$ Teff cells. Mice were sacrificed 5 weeks later and CD4$^+$ T cells from lymph nodes were isolated. CD4$^+$ T cells were activated by TPA/A23187 (50/500 ng ml$^{-1}$), Foxp3$^+$IL-17A$^+$ and Foxp3$^+$IFN-γ$^+$ subpopulations in CD45.2$^+$ tTreg cells (**e**) and CD45.1$^+$ Teff cells (**f**), and Foxp3$^+$ frequency in CD45.2$^+$ tTreg cells (**g**) were determined. PB Pacific blue. Left panels (**e**, **f**), representative staining plots. Right panel **e**, **f**, percentages of Foxp3$^+$IL-17A$^+$ and Foxp3$^+$IFN-γ$^+$ cells. Data are presented as the mean ± SEM, $n = 3$ (*Hif2a$^{f/f}$*) or 7 (*Cd4$^{Cre}$Hif2a$^{f/f}$*) **e–g**. Red circle, *Hif2a$^{f/f}$*; blue square, *Cd4$^{Cre}$Hif2a$^{f/f}$*. *$P = 0.019$ (IL-17A) and 0.024 (IFN-γ) for **e**, $P = 0.0451$ (IL-17A) and 0.328 (IFN-γ) for **f**, $P = 0.0734$ **g** by two-tailed unpaired t-test. Source data are provided as a Source data file.

that HIF-1α protein stability was increased in *Cd4$^{Cre}$Hif2a$^{f/f}$* iTreg cells, relative to that of *Hif2a$^{f/f}$* iTreg cells (Fig. 5g). We further examined if the HIF-2α-specific inhibitor PT2385 (refs. [54,55]) exhibited a similar effect on HIF-1α protein as HIF-2α knockout. Treatment of activating WT CD4$^+$ T cells with PT2385 led to dose-dependent upregulation of HIF-1α (Fig. 5h). PT2385 also increased the level of HIF-1α induction in activated WT iTreg cells (Fig. 5i). Treatment of WT tTreg cells with PT2385 also led to enhanced production of IL-17 (Fig. 5j), similar to our results depicted in Fig. 3b. Levels of the HIF-1α transcriptional targets *Glut1* and *Ccr4* (refs. [56,57]) were significantly higher in *Cd4$^{Cre}$Hif2a$^{f/f}$* tTreg cells (Fig. 5k), along with a weak increase of *Pdk1*, *Srebp1c, and Ccr9*, even though the expression of several other HIF-1α targets was comparable between *Hif2a$^{f/f}$* and *Cd4$^{Cre}$Hif2a$^{f/f}$* tTreg cells (Supplementary Fig. 8). Together, these results illustrate that HIF-2α deficiency enhances HIF-1α expression in T cells and Treg cells, suggesting a possible contribution of elevated HIF-1α levels to the impaired suppressive activity of HIF-2α-KO Treg cells. Notably, *Hif2a* expression was increased in HIF-1α-KO CD4$^+$ T cells and Treg cells (Fig. 5l), suggestive of mutual regulation between HIF-1α and HIF-2α.

**HIF-1α KO restores HIF-2α-KO Treg cell functions.** To elucidate if HIF-1α upregulation destabilized HIF-2α-KO Treg cells, we crossed *Cd4$^{Cre}$Hif2a$^{f/f}$* mice with *Cd4$^{Cre}$Hif1a$^{f/f}$* mice to generate mice from which both HIF-1α and HIF-2α had been deleted. CD45.2$^+$ *Hif2a$^{f/f}$*, *Cd4$^{Cre}$Hif2a$^{f/f}$* and *Cd4$^{Cre}$Hif1a$^{f/f}$Hif2a$^{f/f}$* tTreg cells were adoptively transferred to CD45.1$^+$ RAG-1-KO mice after prior CD45.1$^+$ effector T cell treatment. Additional *Hif1a* deletion restored the capacity of *Cd4$^{Cre}$Hif2a$^{f/f}$* tTreg cells to suppress effector T cell-induced colitis in RAG-1-KO mice, as shown by retention of body weight and repression of colitis score (Fig. 6a, b). Isolation of the transferred tTreg cells (CD45.2) from RAG-1-KO mice illustrated that the elevated expression of IL-17A and IFN-γ in *Cd4$^{Cre}$Hif2a$^{f/f}$* tTreg cells was reduced in *Cd4$^{Cre}$Hif1a$^{f/f}$Hif2a$^{f/f}$* Treg cells (Fig. 6c). By contrast, the proportion of Foxp3$^+$ cells, as determined by flow cytometry, was comparable between recovered *Hif2a$^{f/f}$*, *Cd4$^{Cre}$Hif2a$^{f/f}$* and *Cd4$^{Cre}$Hif1a$^{f/f}$Hif2a$^{f/f}$* tTreg cells (Fig. 6c, d). We also generated *Hif2a$^{f/f}$*, *Cd4$^{Cre}$Hif2a$^{f/f}$* and *Cd4$^{Cre}$Hif1a$^{f/f}$Hif2a$^{f/f}$* iTreg cells, and investigated if additional deletion of HIF-1α corrected the defects of HIF-2α-KO iTreg cells to suppress allergen-induced airway inflammation. *Cd4$^{Cre}$Hif1a$^{f/f}$Hif2a$^{f/f}$* iTreg cells were comparable to *Hif2a$^{f/f}$* iTreg cells in terms of inhibition of ovalbumin-triggered airway resistance, in contrast to *Cd4$^{Cre}$Hif2a$^{f/f}$* iTregs (Fig. 6e). Similarly, HIF-1α-HIF-2α-double knockout (DKO) iTreg cells suppressed allergen-triggered IFN-γ, IL-4, and IL-6 in

BALF as effectively as WT iTreg cells, unlike HIF-2α-KO iTreg cells (Fig. 6f). We examined if iTreg cells could inhibit inflammation when iTreg cells were administered after inducing an allergic response. Whereas HIF-2α-KO iTreg cells were unable to inhibit OVA-induced airway resistance, HIF-1α-HIF-2α-DKO iTreg cells were indistinguishable from WT iTreg cells in their suppression of airway resistance (Supplementary Fig. 9). Therefore, deletion of HIF-1α restored the suppressive activity of HIF-2α-KO Treg cells, suggesting that the impaired function of HIF-2α-KO Treg cells could be partly attributable to a modest elevation of HIF-1α.

**HIF-2α-KO in Treg cells confers resistance to tumor growth.** The immunosuppressive role of Treg cells is well known, which can lead to cancer progression. Destabilization of Treg cells by a deficiency of critical Treg-associated factors such as Nrp-1, SOCS1, EZH2, Eos or CARMA1 protects hosts from syngeneic tumor growth[20,21,31–33]. Recent studies have also revealed that upregulation of HIF-1α induces fragility in Treg cells having anticancer immunity[22,30]. We examined if the modest increase in HIF-1α in HIF-2α-KO Treg cells was sufficient to protect a host from cancer growth. We used *Foxp3$^{Cre}$Hif2a$^{f/f}$* mice in which HIF-2α was only deleted from Treg cells. Inoculation of syngeneic MC38 colon adenocarcinoma led to progressive tumor growth in WT (*Foxp3$^{Cre}$*) mice (Fig. 7a). By contrast, *Foxp3$^{Cre}$Hif2a$^{f/f}$* mice were highly resistant to MC38 growth, with four of the eight mice exhibiting pronounced suppression of tumor growth and the other four mice being completely tumor-free (Fig. 7a). However, *Foxp3$^{Cre}$Hif2a$^{f/f}$* mice were unable to suppress the growth of less immunogenic B16F10 melanoma (Supplementary Fig. 10). We also examined if Treg-selective knockout of HIF-2α interfered with B16F10 metastasis. Intravenous administration of B16F10 resulted in lung metastases in WT mice (Fig. 7b–d). However, Treg-selective deletion of HIF-2α suppressed melanoma metastases, as revealed by overall lung morphology, numbers of tumor nodules and tissue sections (Fig. 7b–d). Therefore, conditional ablation of HIF-2α in Treg cells is sufficient to protect a host from tumor growth.

Studies have shown that administration of inflammatory Treg cells inhibits tumor growth in vivo[20]. We examined if inhibition of HIF-2α in Treg cells also conferred tumor-suppressive activity on Treg cells. Treg cells were treated with PT2385 and then intravenously delivered into mice pre-implanted with MC38 cancer. We found that PT2385-treated Treg cells inhibited the growth of MC38 cancer cells (Fig. 7e). These results suggest that administering HIF-2α-KO Treg cells may be a therapeutically viable approach to suppressing established tumors.

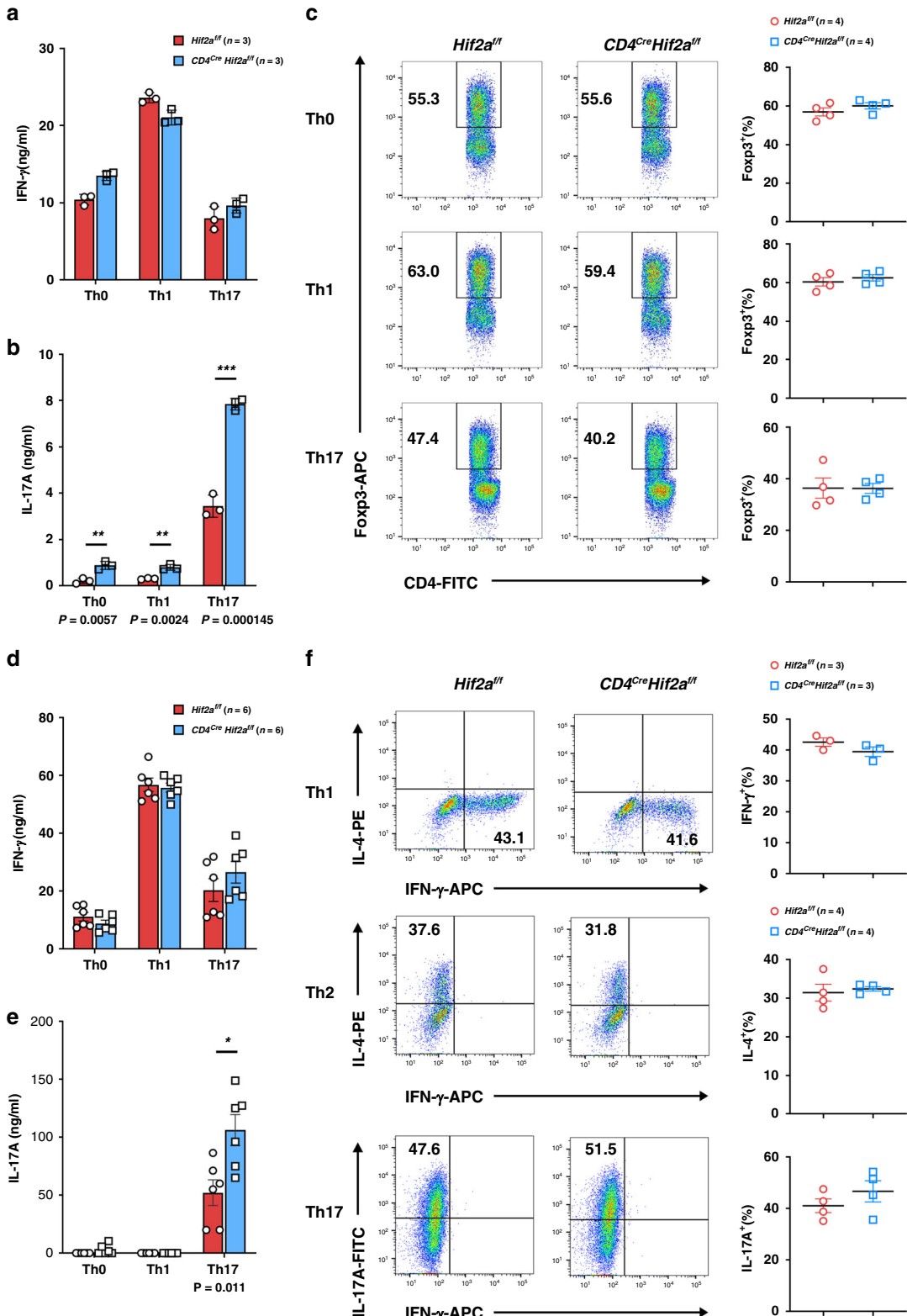

## Discussion

In the present study, we characterize the role of HIF-1α and HIF-2α in the development and function of Treg cells. We find that HIF-1α and HIF-2α exhibit distinct roles in Treg cells and identify an unexpected function of HIF-2α. The role of HIF-1α in Treg cells has been elucidated in many studies, but remains controversial. HIF-1α has been implicated in the expression of Foxp3 due to the presence of a putative HIF-responsive element at the promoter of the *Foxp3* gene[46]. However, HIF-1α binds Foxp3 protein and downregulates Foxp3 (refs. [41,43]), and also tunes up glycolysis to suppress Treg development[42]. Here, we find that HIF-1α knockout does not affect the development or expression of Foxp3 in tTreg cells (Fig. 1a). Our data on differentiation of iTreg cells from HIF-1α-KO CD4+ T cells (Fig. 1c, d) clearly support an inhibitory role of HIF-1α in Foxp3 expression, as previously reported[41,43]. Foxp3 levels (mean fluorescence

**Fig. 3 HIF-2α-KO increases conversion of Treg cells into IL-17A-secreting cells. a–c** Increased transition of HIF-2α-KO tTreg cells into IL-17-expressing cells. *Hif2a*$^{f/f}$ (WT) and *Cd4*$^{Cre}$*Hif2a*$^{f/f}$ tTreg cells were activated with plate-bound anti-CD3/CD28 (4/2 μg ml$^{-1}$) and IL-2 (Th0), with an additional 50 ng/ ml of IL-12 (Th1), or an additional IL-6 (50 ng ml$^{-1}$) plus IL-1 (IL-1α and IL-1β, 20 ng ml$^{-1}$ each) (Th17) for 5 days. IFN-γ **a** and IL-17A **b** levels were measured in supernatants collected from tTreg cells reactivated by TPA/A23187 (20/200 ng ml$^{-1}$) for 16 h. Data **a**, **b** are expressed as mean ± SD, *n* = 3. The frequency of Foxp3$^+$ cells was determined by intracellular staining, and representative plots of four independent experiments are shown in the left panel of **c**. Right panel **c**, data are expressed as mean ± SEM, *n* = 4. Red circle, *Hif2a*$^{f/f}$; blue square, *Cd4*$^{Cre}$*Hif2a*$^{f/f}$. **P = 0.0057 (Th0), 0.0024 (Th1); ***P < 0.001 (Th17) (**b**). **d**, **e** Increased conversion of HIF-2α-KO iTreg cells into IL-17-producing cells. *Hif2a*$^{f/f}$ (WT) and *Cd4*$^{Cre}$*Hif2a*$^{f/f}$ iTreg cells, generated as described in Fig. 1c, were rested for 2 days and then reactivated with plate-bound anti-CD3/CD28 (4/2 μg ml$^{-1}$) under Th1-, Th2-, and Th17- priming conditions as described in **a**. Re-programmed iTreg cells were activated by TPA/A23187, and IFN-γ (**d**) and IL-17A (**e**) production was quantitated. Data are expressed as mean ± SEM, *n* = 6. *P = 0.011 (**e**). **f** Normal HIF-2α-deficient T helper cell differentiation. Naïve T cells (CD4$^+$CD25$^−$CD44$^{lo}$CD62L$^{hi}$) from *Hif2a*$^{f/f}$ and *Cd4*$^{Cre}$*Hif2a*$^{f/f}$ mice were differentiated under either Th1-, Th2-, or Th17-polarizing conditions, as described in Methods, for 5 days. T cells were then re-stimulated with TPA/A23187 (50/500 ng ml$^{-1}$), and expression of IFN-γ, IL-4, and IL-17 was determined by intracellular staining. Numbers represent the percentage of cells positive for IFN-γ, IL-4, or IL-17. Left, representative plots of three independent experiments are shown. Right, data are expressed as mean ± SEM. *n* = 3 (IFN-γ$^+$) or *n* = 4 (IL-4$^+$ or IL-17$^{+)}$. ***P < 0.001. P-value was determined by two-tailed unpaired *t*-test (**a**–**f**). Source data are provided as a Source data file.

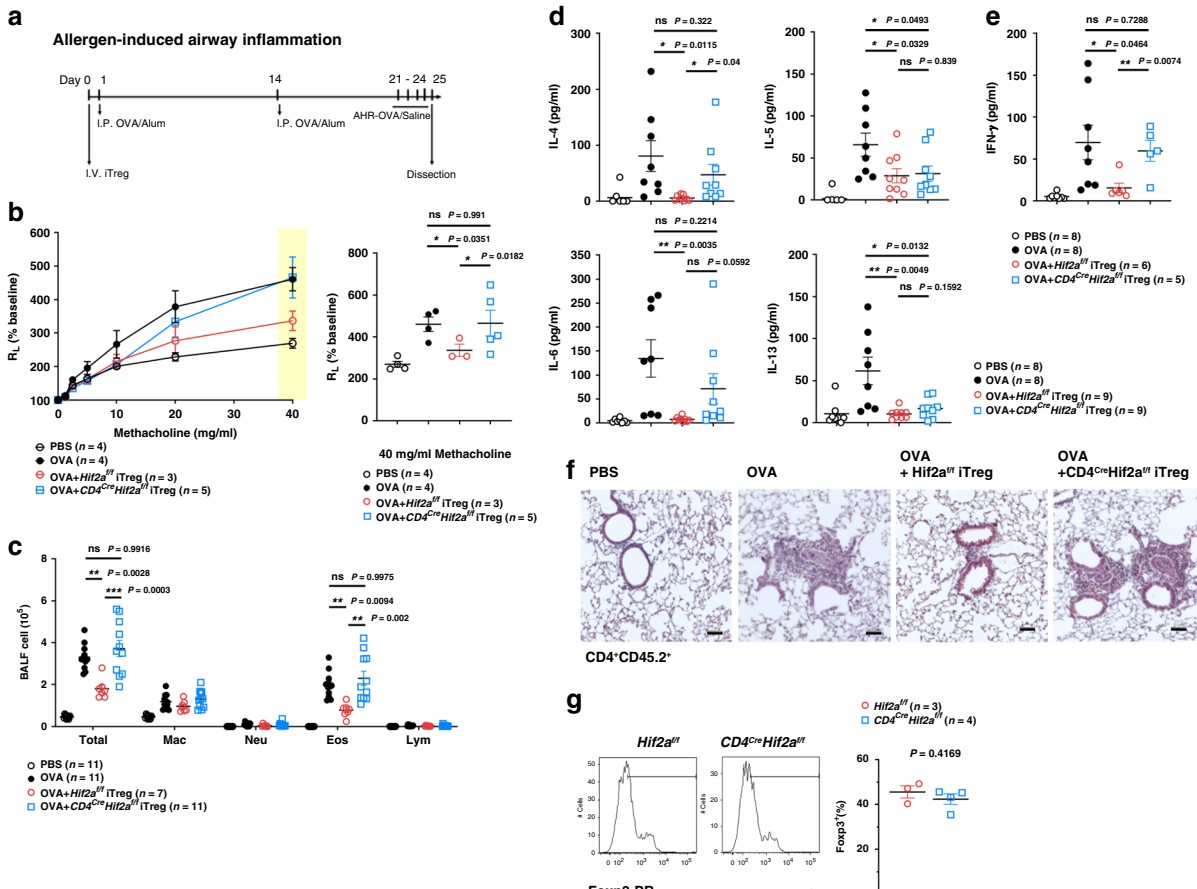

**Fig. 4 HIF-2α-KO iTreg cells do not inhibit allergic inflammation. a** CD45.1$^+$ C57BL/6 mice were administered with CD45.2$^+$ *Hif2a*$^{f/f}$ (WT) and *Cd4*$^{Cre}$*Hif2a*$^{f/f}$ iTreg cells (2.5 × 10$^6$) on day 0, sensitized with ovalbumin (OVA) on days 1 and 14. On day 21, mice were challenged with OVA for four consecutive days and sacrificed on day 25. **b** Diminished ability of HIF-2α-KO iTreg cells to repress airway inflammation. Left panel, lung resistance ($R_L$) in response to methacholine. Right panel, lung resistance at 40 mg ml$^{-1}$ methacholine is expressed as mean ± SEM. Open circle, PBS (*n* = 4); black solid circle, OVA (*n* = 4); red circle, OVA + *Hif2a*$^{f/f}$ iTreg (*n* = 3); blue square, OVA + *Cd4*$^{Cre}$*Hif2a*$^{f/f}$ iTreg (*n* = 5). *P < 0.05, as determined by two-way ANOVA. ns, not significant. **c** Leukocyte population in bronchoalveolar lavage fluid (BALF) from mice in **a**. Numbers of total leukocytes (Total), macrophages (Mac), neutrophils (Neu), eosinophils (Eos), and lymphocytes (Lym) in BALF were determined. Data (cells ml$^{-1}$ in BALF) are expressed as mean ± SEM. PBS, *n* = 11; OVA, *n* = 11; OVA + *Hif2a*$^{f/f}$ iTreg, *n* = 7; OVA + *Cd4*$^{Cre}$*Hif2a*$^{f/f}$ iTreg, *n* = 11. **P < 0.01, ***P < 0.001, as determined by two-way ANOVA with Tukey's multiple comparisons test. ns, not significant. **d**, **e** Levels of IL-4, IL-5, IL-6, IL-13 (d), IFN-γ (e) in BALF were determined. PBS, *n* = 8 **d**, **e**; OVA, *n* = 8 **d**, **e**; OVA + *Hif2a*$^{f/f}$ iTreg, *n* = 9 **d**, 6 (**e**); OVA + *Cd4*$^{Cre}$*Hif2a*$^{f/f}$ iTreg, *n* = 9 **d**, 5 **e**. **f** Representative H&E-stained lung sections are shown. Bar indicates 100 μm. Results were reproduced in six independent experiments. **g** Lung draining lymph nodes were isolated and the percentage of Foxp3$^+$ cells in the CD4$^+$CD45.2$^+$ T cell population was determined· Red circle, *Hif2a*$^{f/f}$ (*n* = 3), blue square, *Cd4*$^{Cre}$*Hif2a*$^{f/f}$ (*n* = 4). Data are presented as mean ± SEM (**d**, **e**, **g**). *P < 0.05, **P < 0.01 (**d**, **e**, **g**), as determined by two-tailed unpaired *t*-test. ns not significant. Source data are provided as a Source data file.

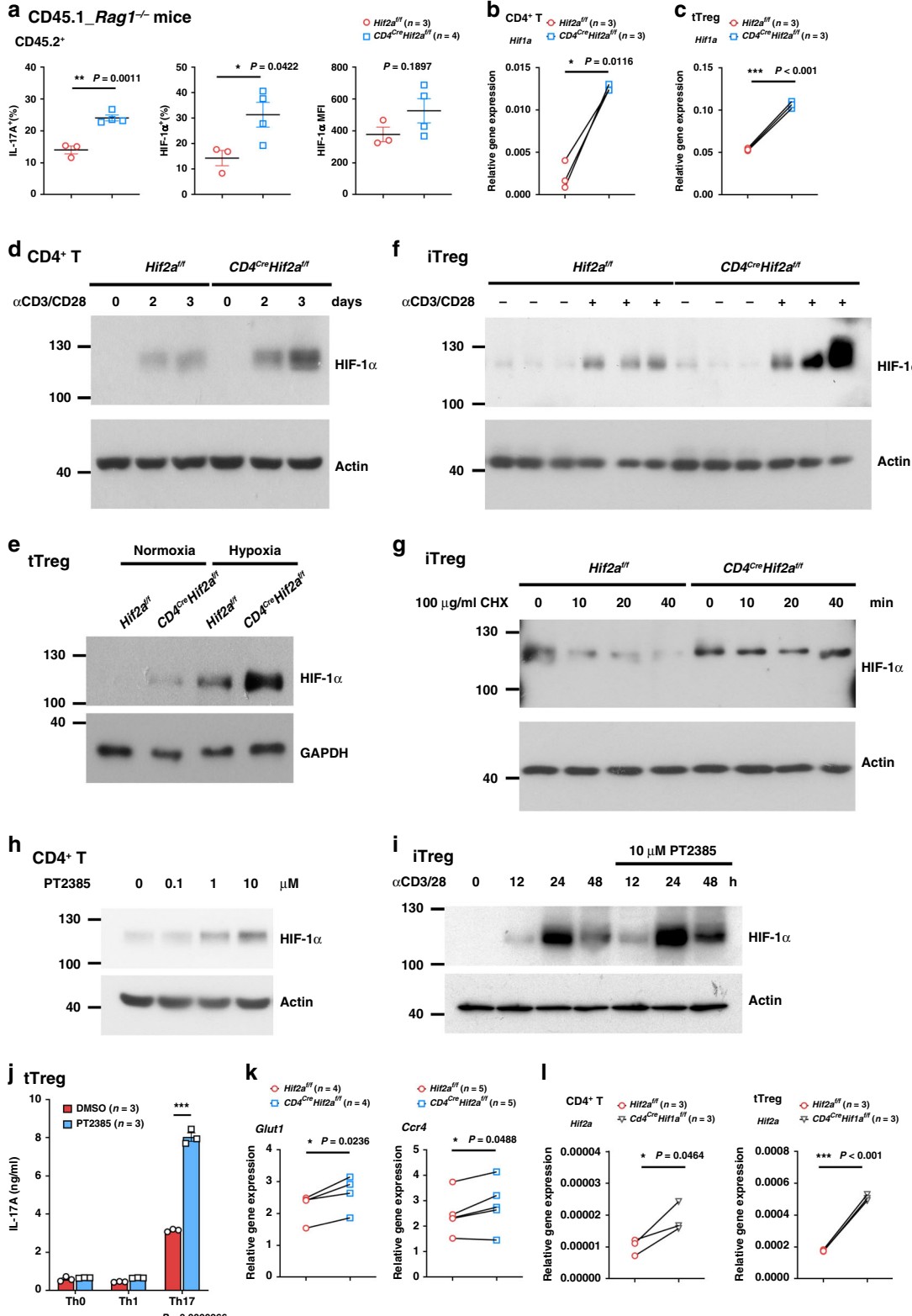

intensity, MFI) in iTreg cells are higher in HIF-1α-KO iTreg cells relative to control when differentiated under normoxic conditions (Fig. 1c). For iTreg cells differentiated under hypoxic conditions, when HIF-1α protein is induced, both the proportion of Foxp3$^+$ cells and Foxp3 MFI are significantly elevated in $Cd4^{Cre}Hif1a^{f/f}$ iTreg cells at higher concentrations of TGF-β (Fig. 1d). Similarly, both those attributes are higher in $Foxp3^{Cre}Hif1a^{f/f}$ iTreg cells developed under hypoxia than for $Foxp3^{Cre}$ and $Foxp3^{Cre}Hif2a^{f/f}$

iTreg cells (Supplementary Fig. 6), indicating that the suppressive effect of HIF-1α persists after Foxp3 induction.

However, the differentiation of WT ($Hif2a^{f/f}$) iTreg cells at suboptimal conditions (0.5 ng ml$^{-1}$ TGF-β) under hypoxia was greater than for $Cd4^{Cre}Hif1a^{f/f}$ and $Cd4^{Cre}Hif2a^{f/f}$ iTreg cells, but not at higher TGF-β concentrations (Fig. 1d, indicated by arrow). This scenario is analogous to a previous report showing that hypoxia increases WT iTreg differentiation in a similar setting

**Fig. 5 HIF-2α-KO cells have high expression of HIF-1α. a** CD45.2$^+$ tTreg cells were transferred to CD45.1$^+$ RAG-1-KO mice together with CD45.1$^+$ Teff. CD45.2$^+$ tTreg cells were re-isolated after 15 days, activated by TPA/A23187 (50/500 ng ml$^{-1}$), and the intracellular HIF-1α, Foxp3, and IL-17 determined. $n = 3$ (*Hif2a$^{f/f}$*), 4 (*Cd4$^{Cre}$Hif2a$^{f/f}$*). Data represent mean ± SEM. *$P = 0.011$ (IL-17A$^+$), 0.0422 (HIF-1α$^+$) by two-tailed unpaired t-test. **b, c** *Hif1a* in *Hif2a$^{f/f}$* and *Cd4$^{Cre}$Hif2a$^{f/f}$* naïve CD4$^+$ T (**b**) and tTreg (**c**) cells was determined by quantitative PCR. $n = 3$. *$P = 0.0116$ **b**, ***$P < 0.001$ **c** by two-tailed paired *t*-test. **d** WT and HIF-2α-KO CD4$^+$ naïve T cells were activated by CD3/CD28 (2/1 μg ml$^{-1}$) and HIF-1α levels were determined. **e, f** *Hif2a$^{f/f}$* and *Cd4$^{Cre}$Hif2a$^{f/f}$* tTreg cells were activated by CD3/CD28 (4/2 μg ml$^{-1}$) for 24 h under normoxic or hypoxic conditions (**e**), or iTreg cells were activated by CD3/CD28 (2/1 μg ml$^{-1}$) for 4 h (**f**), and HIF-1α levels were determined. Each sample **f** represents independent iTreg cells preparation. **g** *Hif2a$^{f/f}$* and *Cd4$^{Cre}$Hif2a$^{f/f}$* iTreg cells were activated as in **f** under hypoxia for 4 h. Cells were then treated with cycloheximide (100 μg ml$^{-1}$) and HIF-1α levels determined. **h, i** WT naïve CD4$^+$ T cells (**h**) were activated by CD3/CD28 (2/1 μg ml$^{-1}$) in the presence of HIF-2α inhibitor PT2385 for 2 days, or activating WT iTreg cells (**i**) were treated with 10 μM PT2385, and HIF-1α levels determined. Results have been reproduced in three (**d, h**) or two **e–i** independent experiments. **j** WT tTreg cells were stimulated with or without 10 μM PT2385, and IL-17 production was quantitated. Data are technical triplicates, the experiment was independently reproduced twice. **k** *Hif2a$^{f/f}$* and *Cd4$^{Cre}$Hif2a$^{f/f}$* tTreg cells were analyzed for expression of *Glut1* and *Ccr4* by quantitative PCR. $n = 4$ (*Glut1*), $n = 5$ (*Ccr4*). *$P = 0.0236$ (*Glut1*) and 0.0488 (*Ccr4*), by two-tailed paired *t*-test. **l** The expression of *Hif2a* in *Hif1a$^{f/f}$* and *Cd4$^{Cre}$Hif1a$^{f/f}$* naïve CD4$^+$ T and tTreg cells was determined. $n = 3$. *$P = 0.0464$, ***$P < 0.001$, by two-tailed paired *t*-test. Source data are provided as a Source data file.

(CD3/CD28, IL-2, and 0.75 ng ml$^{-1}$ TGF-β), which led to the suggestion that hypoxia activates *Foxp3* expression through HIF-1α and the HIF-responsive element on the *Foxp3* promoter[46]. Notably, enhanced hypoxia-mediated WT iTreg differentiation at low TGF-β was not observed in iTreg cells differentiated from *Foxp3$^{Cre}$* T cells, relative to *Foxp3$^{Cre}$Hif1a$^{f/f}$* and *Foxp3$^{Cre}$Hif2a$^{f/f}$* iTreg cells (Supplementary Fig. 6), suggesting that the enhancing effect of hypoxia is limited to *Foxp3* expression as it is no longer detectable after *Foxp3* is expressed. Therefore, hypoxia may promote *Foxp3* expression in vitro, but that effect can only be detected in a narrow timeframe. Together with our finding that HIF-1α mediates suppression of Foxp3 levels at higher TGF-β concentrations under hypoxia (Fig. 1d and Supplementary Fig. 6), we assert that HIF-1α exerts two opposing effects on Foxp3 expression. In iTreg cells differentiated under sub-optimal conditions in which levels of Foxp3 are low, HIF-1α may promote Foxp3 expression. Upon Foxp3 being fully expressed, HIF-1α interacts with Foxp3 protein to downregulate HIF-1α protein expression. Thus, our results may help reconcile the two long-standing opposing views[41,43,46] for how HIF-1α regulates Foxp3 expression. However, in contrast to the impaired ability of HIF-1α-KO tTreg cells to suppress colitis[46], we observed nearly comparable capacity between WT and HIF-1α-KO tTreg cells to inhibit effector T cell-induced colitis (Fig. 2a–d). Therefore, HIF-1α-KO tTreg cells are fully functional in vivo.

Recent studies have revealed that the in vivo inhibitory activity of many different Treg cells is distinct from their in vitro suppressive activities (reviewed in ref. [52]). As discussed above, Foxp3 expression is not affected by HIF-1α deficiency in vivo, but may be HIF-1α-dependent in vitro in particular contexts. It is possible that the difference between the suppressive activity of HIF-1α-KO Treg cells in vitro and in vivo is linked to HIF-1α-dependent in vitro Foxp3 expression, since persistent Foxp3 expression is required for Treg inhibitory activity[5,6]. It is also possible that during an in vitro suppressive analysis comprising only Treg cells, presenting cells and effector T cells, the inhibitory effect is quantitated from the close proximity between Treg cells and the target cells, but such bystander suppression would not be measured in an in vivo suppression assay that involves specific cognation[58]. Further studies are needed to examine these possibilities.

How HIF-2α regulates Treg differentiation and function had been unclear. We found that the in vitro suppressive function of tTreg cells and iTreg cells was not affected by HIF-2α deficiency, but HIF-2α-KO tTreg cells were functionally defective in suppressing effector T cell-induced colitis and in inhibiting airway inflammation. HIF-2α-KO tTreg cells and HIF-2α-KO iTreg cells that were stimulated in vitro secreted higher levels of IL-17, but

not IFN-γ, relative to WT tTreg cells (Fig. 3b, e). By contrast, we observed increased expression of IL-17 and IFN-γ in HIF-2α-KO tTreg cells recovered from RAG-1-KO mice with ongoing colitis (Figs. 2e, 6c). It is known that Treg-specific deletion of a number of genes affects Treg inhibitory function in vivo but *not* in vitro. For example, in mice with Treg-selective deletion of CD28, Ubc13, Helios or Ezh2, impaired Treg in vivo suppressive activity is accompanied by development of spontaneous autoimmune diseases, yet in vitro Treg inhibitory function remained normal[59–62]. Mechanisms involved in the in vivo inactivation of these Treg cells include loss of competitiveness due to a survival disadvantage (CD28, Helios), reduced Treg stability (Ubc13), or destabilization of Foxp3-driven gene expression (Ezh2). Moreover, in vitro co-culture of Treg cells may not fully recapitulate the in vivo inflammatory microenvironments in which Treg cells exert their action. We speculate that the in vivo instability of HIF-2α-KO Treg cells likely contributes to the loss of their suppressive function in vivo but not in vitro. It has also been suggested that proliferating memory-type Treg cells are the main Treg populations responsible for suppression in vivo, but their functions have not been determined in an in vitro suppression assay[52]. Accordingly, HIF-2α is likely involved in the functions of memory-type Treg cells but not in other subsets of Treg cells. Further characterization will help establish the validity of that scenario.

We found that HIF-1α was increased in HIF-2α-KO Treg cells re-isolated from RAG-1-KO mice with colitis (Fig. 5a). HIF-2α deficiency leads to upregulated HIF-1α expression at the stage of both transcription and post-transcription (Fig. 5b–g). Upregulation of HIF-1α is known to promote expression of IL-17 (ref. [41]) and IFN-γ[22,30]. Here, we have further demonstrated that elevated HIF-1α in HIF-2α-KO Treg cells contributes to the instability of HIF-2α-KO Treg cells in vivo, as further knockout of HIF-1α restored the function of HIF-2α-KO tTreg cells and HIF-2α-KO iTreg cells in suppressing effector T cells-induced colitis and allergen-triggered airway inflammation, respectively (Fig. 6). Therefore, the fragility of HIF-2α-KO Treg cells may be partly attributable to upregulated HIF-1α. It should be noted that the degree of elevated HIF-1α expression in HIF-2α-KO Treg cells was modest (Fig. 5a). However, that slight increase of HIF-1α in HIF-2α-KO Treg cells led to elevated expression of *Glut1* (Fig. 5k) and a weak increase of *Pdk1* (Supplementary Fig. 8), accounting for glycolytic activity being unfavorable to Treg development[42]. However, expression of many other HIF-1α targets involved in metabolism, as well as RORγT, were not significantly increased in HIF-2α-KO Treg cells (Supplementary Figs. 7, 8). This outcome explains why our *Foxp3$^{Cre}$Hif2a$^{f/f}$* mice did not recapitulate the profound phenotypes of *Foxp3$^{Cre}$Vhl$^{f/f}$* mice with full-blown

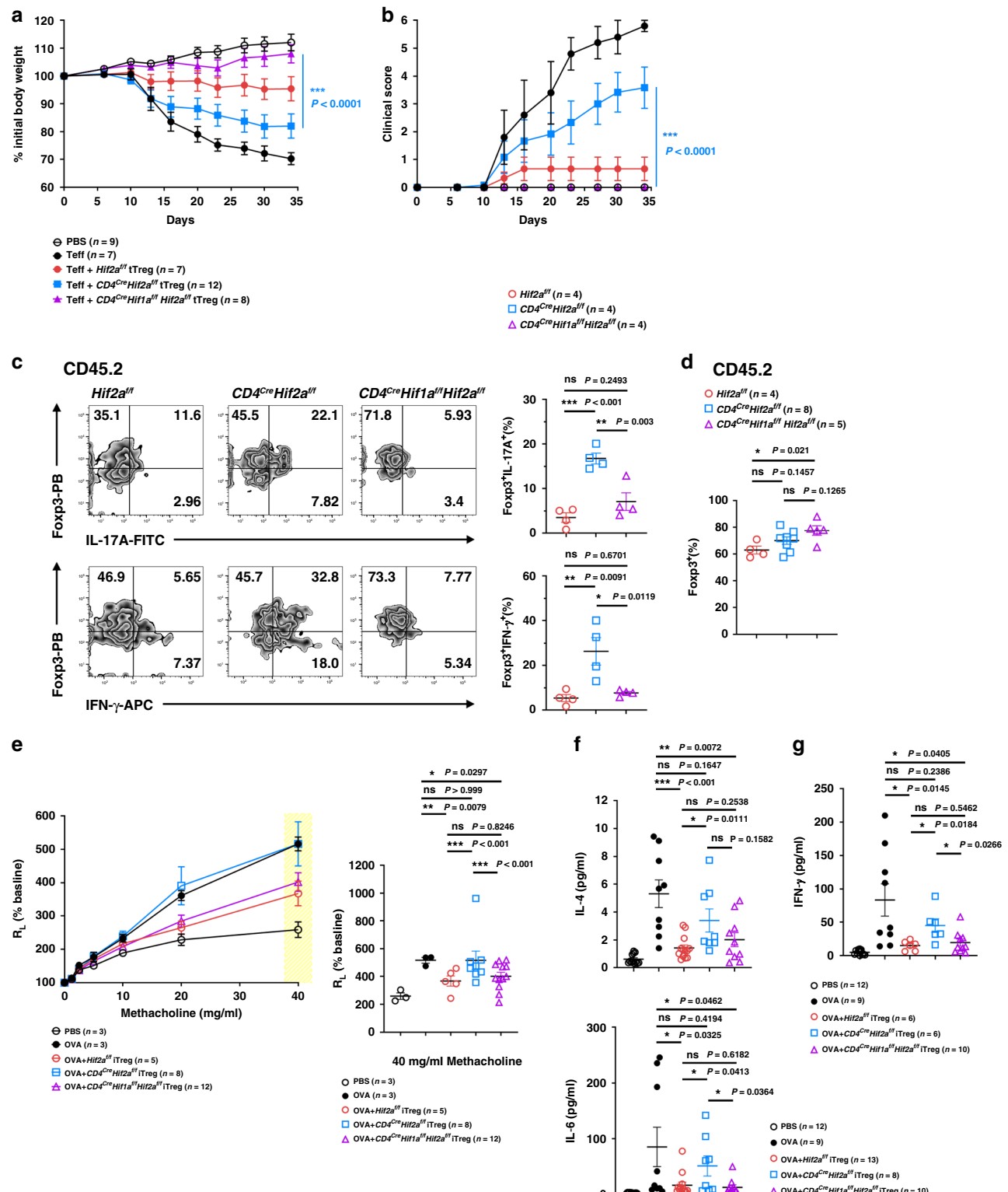

expression of HIF-1α in Treg cells[18], with this latter resulting in massive inflammation, excess IFN-γ production, and premature death.

HIF-1α and HIF-2α regulate the expression of overlapping, as well as a large group of distinct, target genes[36,50,54,55,63,64]. Moreover, activation of HIF target genes is cell type-dependent[65]. In the present study, we have revealed crosstalk between the expression of HIF-1α and HIF-2α in Treg cells. Upregulation of HIF-1α in HIF-2α-KO Treg cells is mediated by both

transcriptional and post-transcriptional mechanisms. *Hif1a* transcript was upregulated in naïve HIF-2α-KO T cells and tTreg cells relative to naïve WT tTreg cells (Fig. 5b, c), indicating that HIF-2α inhibits *Hif1a* transcription. Interestingly, similar upregulation of *Hif2a* was found in HIF-1α-KO T cells and tTreg cells (Fig. 5l). Part of the inhibitory effect of HIF-2α may be attributable to its transcriptional activity given that treatment of WT Treg cells with PT2385, which blocks the dimerization of HIF-2α and HIF-1β, recapitulated some of the phenotypes of HIF-2α-KO

**Fig. 6 HIF-1α KO restores the suppressive activity of HIF-2α-KO Treg cells. a, b** CD45.1[+] RAG-1-KO mice were administered with CD45.1[+] CD4[+]CD25[−] Teff cells ($4 \times 10^5$) with or without CD45.2[+] $Hif2a^{f/f}$, $Cd4^{Cre}Hif2a^{f/f}$ or $Cd4^{Cre} Hif1a^{f/f}Hif2a^{f/f}$ tTreg cells ($1 \times 10^5$). Body weight **a** and colitis scores **b** were assessed. Three out of the 12 RAG-1-KO mice that received HIF-2α-KO tTregs did not lose weight, but their data are included in **a**, **b**. Data represent a combination of three independent experiments. PBS, $n = 9$; Teff, $n = 7$; Teff + $Hif2a^{f/f}$ Treg, $n = 7$; Teff + $Cd4^{Cre}Hif2a^{f/f}$ Treg, $n = 12$; Teff + $Cd4^{Cre}Hif1a^{f/f}Hif2a^{f/f}$ Treg, $n = 8$. ****$P < 0.0001$, as determined by repeat measures two-way ANOVA with Tukey's multiple comparisons test, with the Greenhouse-Geisser correction. **c, d** Mice from **a** were sacrificed 5 weeks later and CD45.2[+] CD4[+] T cells were isolated, re-stimulated with TPA/A23187, and then expression of Foxp3, IFN-γ and IL-17A was determined. (**c**, left), representative plots; (**c**, right), percentages of Foxp3[+]IL-17A[+] and Foxp3[+]IFN-γ[+] cells; (**d**), Foxp3[+] fraction among the CD4[+]CD45.2[+] T cell population. Data are presented as mean ± SEM. $Hif2a^{f/f}$, n = 4 **c**, **d**; $Cd4^{Cre}Hif2a^{f/f}$, n = 4 **c**, 8 **d**; $Cd4^{Cre}Hif1a^{f/f}Hif2a^{f/f}$, n = 4 (**c**), 8 (**d**). **e–g** B6 mice were treated with $Hif2a^{f/f}$, $Cd4^{Cre}Hif2a^{f/f}$ and $Cd4^{Cre}Hif1a^{f/f}Hif2a^{f/f}$ iTreg cells. $R_L$ in response to methacholine was measured (**e**, left) and data is represented as mean ± SEM (e, right). Amounts of IL-4, IL-6 (**f**), and IFN-γ **g** in the BALF were determined. PBS, $n = 3$ (**e**),12 (**f, g**); OVA, $n = 3$ (**e**), 9 (**f, g**); OVA + $Hif2a^{f/f}$ iTreg, $n = 5$ (**e**), 13 (**f**), 6 (**g**); OVA + $Cd4^{Cre}Hif2a^{f/f}$ iTreg, $n = 8$ (**e**), 8 (**f**), 6 (**g**); OVA + $Cd4^{Cre}Hif1a^{f/f}Hif2a^{f/f}$ iTreg, $n = 12$ (**e**), 10 (**f, g**). Data are presented as mean ± SEM (**f, g**). *$P < 0.05$, **$P < 0.01$, ***$P < 0.001$ (**c–g**), as determined by two-tailed unpaired $t$-test (**c, d, f, g**), or two-way ANOVA with Tukey's multiple comparisons test **e**. Source data are provided as a Source data file.

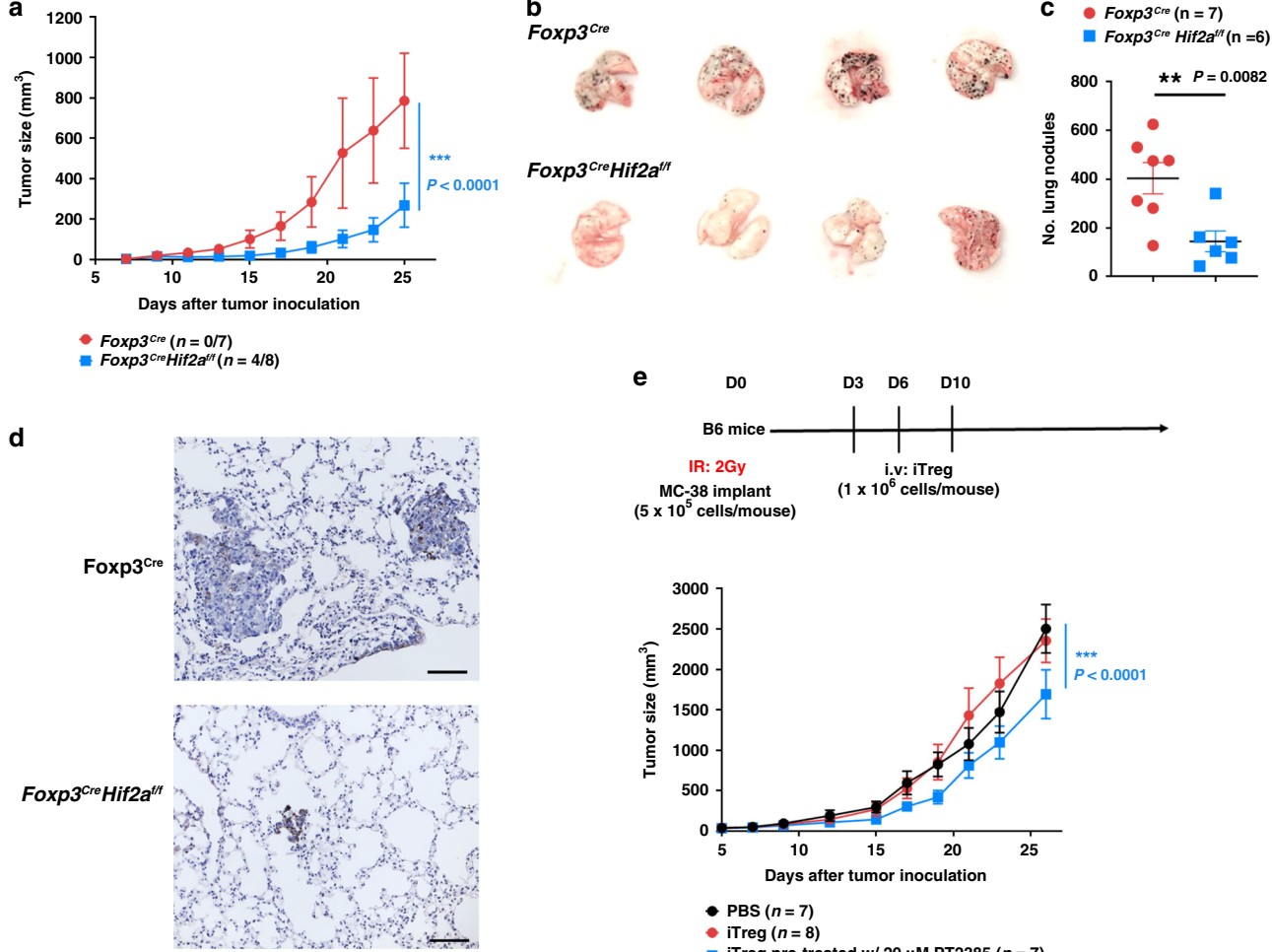

**Fig. 7 Treg-specific KO of HIF-2α confers resistance to tumor growth. a** Treg-conditional deletion of $Hif2a$ inhibits MC38 growth. MC38 colon adenocarcinoma cells ($1 \times 10^5$) were subcutaneously inoculated into WT ($Foxp3^{Cre}$) or $Foxp3^{Cre}Hif2a^{f/f}$ mice, with $Foxp3^{Cre}$ being littermate controls of $Foxp3^{Cre}Hif2a^{f/f}$ mice. Tumor growth was assessed every 2 days. Plotted values represent the mean and SEM of mice with a tumor. Numbers in brackets of legend represent tumor-free mice in each group. Red circle, $Foxp3^{Cre}$ ($n = 7$), blue square, $Foxp3^{Cre}Hif2a^{f/f}$ ($n = 8$). ****$P < 0.0001$. **b–d** $Foxp3^{Cre}Hif2a^{f/f}$ mice are resistant to B16F10 metastases. B16F10 melanoma cells ($2 \times 10^5$) were intravenously administered into $Foxp3^{Cre}$ and $Foxp3^{Cre}Hif2a^{f/f}$ mice, with $Foxp3^{Cre}$ mice being littermates of $Foxp3^{Cre}Hif2a^{f/f}$ mice. At day 12 post-implantation, lungs were isolated **b** and the numbers of black nodules in lungs were quantified **c**. Red circle, $Foxp3^{Cre}$ ($n = 7$), blue square, $Foxp3^{Cre}Hif2a^{f/f}$ ($n = 6$) **c**. **$P = 0.0082$. Lung metastases were examined by H&E staining **d**. Bar indicates 100 µm. **e** Inhibition of MC38 growth by HIF-2α-downregulated Treg cells. Upper panel shows experimental setup. C57BL/6 male mice (6–8 weeks-old) were transplanted with $5 \times 10^5$ MC38 cells after 2 Gy irradiation on day 0. PBS, iTreg cells ($1 \times 10^6$), or iTreg cells treated with PT2385 were adoptively transferred into MC38-bearing mice on days 3, 6, and 10. Tumor growth was assessed and is represented as mean tumor volume ± SEM. Black circle, PBS ($n = 7$); red circle, iTreg ($n = 8$); blue square, PT2385-treated iTreg ($n = 7$). ****$P < 0.0001$. $P$-value was determined by repeat measures two-way ANOVA with Tukey's multiple comparisons test, with the Greenhouse-Geisser correction **a, e** or two-tailed unpaired $t$-test (**c**). Source data are provided as a Source data file.

tTreg cells (Fig. 5j). HIF-1α protein is also subjected to various modifications including phosphorylation, sumoylation, deacetylation and ubiquitination that regulate its stability[65]. Absence of HIF-2α led to increased HIF-1α protein stability in iTreg cells (Fig. 5g). HIF-2α regulates over 1000 genes that are HIF-1α-independent[64], so the modulation of HIF-1α protein stability by HIF-2α deficiency could be mediated by HIF-2α target gene products, but not necessarily by HIF-2α itself. Further works are required to map out exactly how HIF-2α modulates HIF-1α expression and stability.

The inhibitory activities of Treg cells contribute to immuno-suppressive microenvironments in tumors[26,27]. Depletion or destabilization of Treg cells represents a current approach to re-activating anticancer immunity[29]. HIF-1α promotes Foxp3 instability and degradation, induces IL-17 expression, stimulates IFN-γ production, and drives Treg fragility[22,30,41,43,44]. Through these activities, HIF-1α expression in Treg cells confers anticancer immunity, which protects hosts from tumor growth. Selectively increased HIF-1α expression in Treg cells could therefore be considered a therapeutic approach to treating cancer because it would destabilize Treg cells and convert them into inflammatory T cells. However, high HIF-1α expression in T cells or Treg cells would likely induce lethal autoimmunity in the host[18,30]. In the present study, we found that a moderate increase of HIF-1α expression in HIF-2α-KO Treg cells did not result in auto-immunity in $Cd4^{Cre}Hif2a^{f/f}$ or $Foxp3^{Cre}Hif2a^{f/f}$ mice. Interestingly, that moderate HIF-1α expression level in Treg cells was sufficient to protect the host from MC38 colon adenocarcinoma and metastatic B16F10 invasion (Fig. 7). We have further demonstrated that the HIF-2α inhibitor PT2385 promotes upregulation of HIF-1α in T cells (Fig. 5h, i), and that adoptive transfer of PT2385-treated Treg cells was able to inhibit growth of MC38 (Fig. 7e). Therefore, our results suggest a potential anti-cancer therapeutic application of moderately increasing HIF-1α expression in Treg cells to downregulate HIF-2α. HIF-2α has been shown to be overexpressed and is the oncogenic driver of clear cell renal cell carcinoma (ccRCC), and targeting of HIF-2α by HIF-2α-specific inhibitors represents a promising therapeutic strategy for treating ccRCC[54,55]. Our results may indicate that destabilization of Treg cells by HIF-2α inhibitors contributes to the anticancer effect by upregulating HIF-1α. Further experiments will help determine the exact mechanism by which re-programming of Treg cells by HIF-2α-specific inhibitors can aid in the treatment of cancers.

In summary, we have demonstrated crosstalk between the expression of HIF-2α and HIF-1α in Treg cells, which contributes to an unexpected role of HIF-2α in Treg stability. Therapeutic applications of *ex vivo*-expanded Treg cells have been explored extensively[23–25], including manipulation of specific gene expression in Treg cells. HIF-1α and HIF-2α are known for their prominent roles in autoimmune diseases, immunity and cancer, and reagents to modulate their expression are being examined in clinical trials or are being developed. Our results indicate that controlled expression of HIF-2α and HIF-1α in Treg cells could be used to maintain the suppressive activity of Treg cells or to generate fragile Treg cells. Further studies should pursue this research direction to optimize the therapeutic potential of Treg cells in different applications.

## Methods

**Reagents.** Recombinant mouse IL-12 (419-ML), mouse IL-1α (400-ML), mouse IL-1β (401-ML), and anti-mouse/human HIF-1α-Alexa Fluor 700 (241812, 1:100) were purchased from R&D systems (Minneapolis, MN). Antibodies against GAPDH (6C5, 1:6000) was obtained from Santa Cruz Biotechnology (Santa Cruz, CA). Antibodies against HIF-1α (10006421, 1:2000) were purchased from Cayman Chemical (Ann Arbor, MI). Antibodies against actin (MAB1501, C4, 1:4000) were purchased from Merck Millipore (Billerica, MA). Recombinant mouse IL-4

(PMC0045), mouse IL-6 (RMIL6I), mouse IL-2 (RP-8608), anti-CD4-FITC (GK1.5 and RM4-5, 1:500), anti-CD44-PE-Cy7 (IM7, 1:1000), anti-CTLA-4-PE (clone UC10-4B9, 1:250), anti-FR4-PE (clone eBio12A5, 1:250), anti-Foxp3-Pacific blue or APC (FJK-16S, 1:200), anti-LAG-3-PE (eBioC9B7W, 1:250), anti-CD45.1-PE (A20, 1:500), and anti-CD45.2-PE-Cy7 (104, 1:500) were purchased from Thermo Fisher (Madison, WI). Recombinant mouse IL-12 (577002), anti-IL-2 (JES6-1A12), anti-IL-4 (11B11), anti-IL-12 (C17.8), anti-IFN-γ (R4-6A2), anti-IL-4-PE (11B11, 1:250), anti-CD8-APC (53-6.7, 1:500), anti-CD62L-APC (MEL-14, 1:500), anti-CD25-FITC (PC61, 1:1000), anti-CD25-PE (PC61, 1:1000), anti-GITR-PE (YGITR 765, 1:250), anti-IFN-γ-APC (XMG1.2, 1:250), anti-IL-17-FITC (TC11-18H10.1, 1:250), anti-mouse/human Helios-PE (22F6, 1:250), and anti-mouse CD304-Bril-liant Violet 421 (Neutrophilin-1, 3E12, 1:500) were purchased from BioLegend (San Diego, CA). Anti-CD3 (2C11) and anti-CD28 (37.51) were purchased from BioXCell (West Lebanon, NH), and anti-mouse CD4 (RL172.4) was purified in our laboratory. ELISA kits for mouse IFN-γ (88-7314-88), mouse IL-17A (88-7371-88), mouse IL-10 (88-7105-88), and mouse TGF-β (88-8350-88) were purchased from Thermo Fisher. Ovalbumin (A2512), 12-O-Tetradecanoylphorbol 13-acetate (TPA, also known as PMA, 79346) and calcium ionophore A23187 (C7522) were obtained from Sigma-Aldrich (St. Louis, MO).

**Cell lines.** B16F10 melanoma cells were obtained from ATCC (CRL-6475, Man-assas, VA). MC38 colon adenocarcinoma cells were purchased from Kerafast (CVCL_B288, Boston, MA). MC38 and B16F10 cells were cultured in DMEM (Life Technologies, Carlsbad, CA) supplemented with 10% FCS (Life Technologies), 1% Penicillin-Streptomycin, 1% L-Glutamine, and 1% HEPES.

**Mice.** $Cd4^{Cre}$ mice were obtained from Taconic Farms. $Rag1^{−/−}$ (RAG-1-KO) mice, $Hif1a^{tm3Rsjo/J}$ ($Hif1a^{f/f}$) mice, $Epas1^{tm1Mcs/J}$ ($Hif2a^{f/f}$) mice, and NOD/ShiLt-Tg ($Foxp3$-EGFP/cre)1cJbs/J mice (also known as NOD.$Foxp3^{Cre}$) were obtained from Jackson Laboratories. NOD.$Foxp3^{Cre}$ mice were back-crossed to C57BL/6 mice for 12 generations ($Foxp3^{Cre}$)[44]. $Cd4^{Cre}$ mice were bred with $Hif1a^{f/f}$ or $Hif2a^{f/f}$ for T cell-specific deletion of $Hif1a$ or $Hif2a$, respectively. $Foxp3^{Cre}$ mice were crossed with $Hif1a^{f/f}$ or $Hif2a^{f/f}$ to produce mice with Treg-specific deletion of $Hif1a$ or $Hif2a$. Mice were housed in SPF animal facility of Institute of Molecular Biology (Academia Sinica), with ambient temperature at 21 °C, humidity of 55%, dark/light cycle of 10 h/14 h, and air exchange rate of 12–15 times per hour. In vivo experiments were conducted on sex-matched male and female mice. Both male and female mice were used for in vitro and in vivo experiments. All mice used in our study were 8–10 weeks old. All mouse experimental protocols were approved by the Institutional Animal Care and Utilization Committee of Academia Sinica.

**T cell activation.** Single-cell suspensions were prepared from lymph nodes or spleen, and immunoglobulin panning was used to isolate total T cells. Specific T cell populations were sorted on a FACSAria II SORP system (BD Biosciences, San Jose, CA), and purified T cells were cultured in complete RPMI 1640 medium. Purified CD4+ T cells were stimulated with plate-bound anti-CD3 and anti-CD28 for 24 h to detect IL-2 production and for 48 h to measure proliferation and IFN-γ levels (Thermo Fisher). T cell proliferation was measured by thymidine incor-poration using a direct beta-counter (MATRIX 96, Packard Instruments, Meriden, CT).

**Preparation and characterization of Treg cells.** CD4+CD25+ thymus-derived regulatory T (tTreg) cells were purified from peripheral lymph nodes and spleens by sorting on a FACSAria II SORP system. Intracellular staining of Helios and Foxp3 was performed on paraformaldehyde-fixed splenocytes pre-labelled for CD4 and CD25 using the Foxp3/Transcription Factor Staining Buffer Set (Thermo Fisher Scientific Inc., Madison). Neuropilin-1 (Nrp-1) staining was conducted with anti-Nrp-1. Naïve IL-10 and TGF-β cytokine secretion from tTreg cells was esti-mated by ELISA as per the manufacturer's instructions (Thermo Fisher) upon stimulating tTreg cells with immobilized anti-CD3 (4 μg ml$^{-1}$) and anti-CD28 (2 μg ml$^{-1}$) for 48–96 h. Gating and sorting strategies for tTreg cells purification used in in vitro and in vivo functional analyses are described in Supplementary Fig. 11a.

For induced Treg (iTreg) differentiation, FACS-sorted naïve T cells were stimulated with immobilized anti-CD3 (2 μg ml$^{-1}$) and anti-CD28 (1 μg ml$^{-1}$) in the presence of TGF-β (5 ng ml$^{-1}$) and IL-2 (20 μg ml$^{-1}$) for 5 days. CD4+CD25+ cells were collected after 5 days of differentiation and used for in vitro and in vivo suppression analysis. Respective gating and sorting strategies are described in Supplementary Fig. 11b.

**In vitro Treg cell suppression assay.** CD4+CD25− effector T (Teff) cells (1 × 10$^5$) were co-cultured with tTreg cells or iTreg cells in the presence of antigen-presenting cells (APC, mitomycin C-treated T-depleted splenic cells, 3 × 10$^5$) and 1 μg ml$^{-1}$ soluble anti-CD3 in 96-well U-bottom plates. The ratios of Treg cells to Teff cells were 0:1, 1:1, 1:2, 1:4, or 1:8. After 72 h, CD4+CD25− Teff cell pro-liferation was measured by thymidine incorporation, and IL-2 and IFN-γ levels were detected by ELISA in supernatants after 40 h of culture.

**In vivo tTreg cell suppression assay**. CD4$^+$CD25$^-$Teff cells ($4 \times 10^5$) from CD45.1$^+$ mice were administered with or without $1 \times 10^5$ CD4$^+$CD25$^+$ tTreg cells from 8 to 10 week-old CD45.2$^+$ mice by intraperitoneal injection into sex matched 8-10 week-old CD45.1$^+$ RAG-1-KO mice. We scored body wasting as follows: 0, no wasting; 1 for up to 10 % initial body weight loss; 2 for 10-20 % initial body weight loss; and 3 for >20% initial body weight loss. Rectal prolapse was scored as 0 for absence and 1 for presence. Diarrhea was scored as 0 for absence (solid stool), 1 for soft stool, and 2 for watery/bloody stool. Body weight and score was assessed every 3 days. Five weeks after T-cell transfer, mouse colons were removed, cleaned using PBS, fixed in 4% paraformaldehyde, embedded in paraffin, and then sectioned and stained with H&E. CD45.2$^+$ T cells (representing transferred Treg cells) from lymph nodes and spleen were isolated from CD45.1$^+$ RAG-1-KO mice, and reactivated by TPA/A23187 for 6 h and stained intracellularly for expression of Foxp3, IFN-γ, and IL-17A. Respective gating strategies are described in Supplementary Fig. 12.

**In vivo iTreg suppressive assay**. WT mice were intravenously injected with $2.5 \times 10^6$ iTreg cells or PBS on day 0, and then sensitized by intraperitoneal injections of 50 μg ovalbumin (OVA) emulsified in 2 mg of aluminum hydroxide (77161, Pierce Chemical Co., Rockford, IL) on day 1 and day 14. Three weeks later, mice were challenged by inhalation (30 min daily for four consecutive days) of aerosolized 1% OVA in saline (0.9% NaCl) generated by a Nebulizer 646 in an aerosol therapy system (DeVilbliss). To evaluate lung function, an allergy-induced airway hyper-reactivity (AHR) assay was performed. Briefly, mice were anesthetized with pentobarbital (P3761, Sigma-Aldrich) at 100 mg/kg body weight after the last OVA challenge. Anesthetized mice were tracheotomized, intubated, and mechanically ventilated at a tidal volume of 0.20 mL and a frequency of 150 breaths/min. Resistive index (RI) values were recorded at baseline and following a 10 s exposure to an increasing concentration of methacholine (A2251, Sigma-Aldrich) (0, 6.125, 12.5, 25, and 50 mg ml$^{-1}$) through the FinePointe RC System (Buxco Research Systems, Wilmington, NC). Immediately after AHR measurement, bronchoalveolar lavage fluid (BALF) was collected by three instillations of 1 mL cold saline, and amounts of IL-4, IL-5, IL-6, IL-13 and IFN-γ in BALF from the first collection were measured by cytometric bead assay (BD Biosciences, San Jose, CA). The relative number of different types of leukocytes in BALF was determined from differentials based on 200 cells by Differential Quik III Staining (Dade Behring, Deerfield, Ill). Results are given as cells ml$^{-1}$ in BALF.

**Treg cell conversion experiment**. tTreg and iTreg cells were stimulated by plate-bound anti-CD3 (4 μg ml$^{-1}$) plus anti-CD28 (2 μg ml$^{-1}$) and in the presence of IL-2 (20 ng ml$^{-1}$) with or without IL-12 (50 ng ml$^{-1}$), or IL-6 (50 ng ml$^{-1}$) plus IL-1 (IL-1α and IL-1β, 20 ng ml$^{-1}$ each). These cells were harvested after 5 days and Foxp3 expression was determined using an LSRII-18P flow cytometry system (BD Biosciences). IFN-γ and IL-17A levels were measured by ELISA in supernatants collected from Treg cells reactivated using TPA (20 ng ml$^{-1}$) and A23187 (200 ng ml$^{-1}$) for 16 h.

**T helper cell differentiation**. Sorted CD4$^+$CD25$^-$CD44$^{lo}$CD62L$^{hi}$ naïve T cells ($1 \times 10^6$ ml$^{-1}$) were stimulated with plate-bound anti-CD3 (2 μg ml$^{-1}$) and anti-CD28 (1 μg ml$^{-1}$) in the presence of 20 ng ml$^{-1}$ mouse IL-12 and 10 μg ml$^{-1}$ anti-mouse IL-4 antibody for Th1 differentiation. For Th2 differentiation, mouse IL-4 (20 ng ml$^{-1}$), anti-IL-12 (10 μg ml$^{-1}$) and anti-IFN-γ antibody (10 μg ml$^{-1}$) were added. Cultures were also supplemented with 20 ng/mL IL-2 during Th1 and Th2 differentiation. For Th17 differentiation, sorted naïve T cells ($2 \times 10^6$ ml$^{-1}$) were stimulated with immobilized anti-CD3 (4 μg ml$^{-1}$) and anti-CD28 (2 μg ml$^{-1}$) supplemented with 30 ng ml$^{-1}$ mouse IL-6, 5 ng ml$^{-1}$ TGF-β, and 10 μg ml$^{-1}$ each of anti-mouse IL-2, anti-mouse IL-4, anti-mouse IL-12, and anti-IFN-γ antibodies. Cells were collected after 5 days of differentiation and stimulated for 5 h with PMA (50 ng ml$^{-1}$) and ionomycin (500 ng ml$^{-1}$) in the presence of monensin (2 nM). Expression of IFN-γ, IL-4, and IL-17 was measured by intracellular staining. The priming of Treg cells into Th0-, Th1- and Th17-like cells is described in Fig. 3 legend.

**Quantitative PCR**. Total RNA from CD4$^+$ T or tTreg cells was collected using a GENEzol triRNA Pure Kit (Geneaid, Taiwan). The eluted RNA was reverse-transcribed using SuperScript III Reverse Transcriptase (Thermo Fisher). cDNA was synthesized from 1 μg of RNA from each sample and analyzed for expression of HIF-1α, HIF-2α or HIF-1α target genes on a LightCycler 480 Real-Time PCR System using software v1.5.1.621.5.1.62 (Roche, Germany). The primers used are listed in Supplementary Table 1. The PCR protocol was 95 °C for 10 min, followed by 45 cycles of 95 °C for 10 s, 60 °C annealing for 10 s, and 72 °C extension for 8 s. Samples were normalized to *Actin* expression.

**Cell lysates and immunoblots**. To prepare whole-cell lysates, cells were lysed by whole-cell extract buffer (25 mM HEPES pH 7.7, 300 mM NaCl, 0.1% Triton X-100, 1.5 mM MgCl$_2$, 0.2 mM EDTA, 0.1 mM Na$_3$VO$_4$, 50 mM NaF, 0.5 mM dithiothreitol and 10% glycerol) on ice for 30 min. Debris was removed by centrifuging on an Eppendorf microfuge (13,200 r.p.m., 20 min, 4 °C). Protein concentrations in the supernatants were quantitated by Bio-Rad protein assay. Protein

samples were resolved by SDS polyacrylamide gel electrophoresis, before being transferred to PVDF membrane in transfer buffer (30 mM Tris, 250 mM glycine, 1 mM EDTA, 20% methanol) at room temperature for 1.5 h at 400 mA. The membrane was blocked with blocking buffer (5% non-fat milk in 10 mM Tris–HCl pH 8.0, 150 mM NaCl, 0.05% Tween 20) at room temperature for 1 h. The treated membrane was incubated with primary antibodies overnight at 4 °C, washed, and incubated with horseradish peroxidase-conjugated secondary antibodies at room temperature for 1.5 h. The membrane was developed using an ECL Western blotting detection kit. Luminescence was detected by X-ray film. Western blot images have been cropped for presentation. Full size images are presented in Supplementary Fig. 13.

**Implantation of tumor cells**. B16F10 melanoma cells were obtained from ATCC (CRL-6475, Manassas, VA). MC38 colon adenocarcinoma cells were purchased from Kerafast (CVCL_B288, Boston, MA). MC38 and B16F10 cells were cultured in DMEM (Life Technologies, Carlsbad, CA) supplemented with 10% FCS (Life Technologies), 1% Penicillin-Streptomycin, 1% L-Glutamine, and 1% HEPES. A total of $1 \times 10^5$ tumor cells was suspended in 100 μl PBS and inoculated subcutaneously into the right hind flank of WT (*Foxp3$^{Cre}$*) or *Foxp3$^{Cre}$Hif2a$^{f/f}$* mice. Tumor size was determined by measuring tumor length and width, using volume as readout. Volumes (V) were calculated using the equation: $V = L \times W^2/2$, where L is the long diameter and W is the short diameter. To evaluate the effect of HIF-2α-suppressed iTreg cells, C57BL/6 mice were lightly irradiated (2 Gy), followed by subcutaneous transplantation with $5 \times 10^5$ MC38 cells. Sorted CD4$^+$CD25$^+$ iTreg cells were untreated or treated with 20 μM of the HIF-2α-specific inhibitor PT2385 (HY-12867, Medchem Express, Monmouth Junction, NJ) for 16 h and then adoptively transferred into mice on days 3, 6 and 10 after tumor implantation.

**Evaluation of B16F10 lung metastasis**. Lung metastases were established in WT or *Foxp3$^{Cre}$Hif2a$^{f/f}$* mice by injecting $2 \times 10^5$ B16F10 melanoma cells in 100 μL PBS via the tail vein. On day 12, the mice were euthanized and then perfused with PBS. Numbers of tumor nodules were counted. Lungs were then fixed in 4% paraformaldehyde followed by 70% alcohol for histologic evaluation. Lung sections were stained with hematoxylin and eosin (H&E) to visualize lung architecture and malignant nodules.

**Statistics**. Lung inflammation experiments and colitis histology were blinded. Other data in this study were collected randomly but not blind, several experiments were confirmed independently by a second person. No data were excluded from this study. Our data generally met the assumptions of the statistical tests applied (i.e. normal distributions). GraphPad Prism 5 and Microsoft Office Excel 2016 were used for data analysis. Flow cytometry data were collected using FACSdiva software version 8.0.2 (BD Biosciences), and were analyzed on FlowJo version 10 (BD Biosciences). Unpaired two-tailed Student $t$-tests and paired two-tailed Student $t$-tests were used to compare results between two groups. Data are presented as mean with standard deviation (SD) or standard error of the mean (SEM), as indicated in the figure legends. Weight loss and clinical score in CD4$^+$ T cell–mediated colitis in mice were analyzed by repeat measures two-way ANOVA with Tukey's multiple comparisons test, with the Greenhouse-Geisser correction. Changes in lung resistance (R$_L$) in response to increasing doses of methacholine and cell composition of BALF were analyzed by two-way ANOVA with Tukey's multiple comparisons test. $P$ values < 0.05 were considered significant.

**Reporting summary**. Further information on research design is available in the Nature Research Reporting Summary linked to this article.

## Data availability
Source data are provided with this paper.

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

## Acknowledgements

We thank Yamin Lin and the Institute of Molecular Biology Academia Sinica (IMB) FACS Core for cell sorting, and Dr. John O'Brien for editing the paper. This work was

supported by grant MOST 107-2321-B-001-031 from the Ministry of Science and Education, Taiwan, R.O.C.

## Author contributions

T.S.H., Y.L.L., Y.A.W., S.T.M., P.Y.C., A.C.Y.L., H.Y.P., and Y.J.C. performed experiments; T.S.H. and M.Z.L. wrote the paper; and M.Z.L. conceived, designed, and supervised the research.

## Competing interests

The authors declare no competing interests.
