## [Peer Review File · Nature Communications]

Reviewers' comments:

Reviewer #1 (Remarks to the Author):

Overall this is a well written paper and the work clearly done. The paper though lacks originality.

The role of HIF-2 alpha has been described in the Triner paper reference (44) and other references, for example (Singh et al 2016). and others.

Although some new aspects of the complex functionalities are described I think that the paper would be better suited for more general immunology journal.

Reviewer #2 (Remarks to the Author):

The manuscript by Hsu, et al., reports on the yet unexplored importance of HIF-2alpha in Regulatory T cells (Tregs). In this study the authors utilize mice with Treg-specific HIF-2 and HIF-1 alpha deficiencies in a number of in vivo disease models (colitis, airway inflammation, cancer), as well as in vitro models of Treg differentiation and function. In their in vivo studies they use flow cytometry and ex vivo functional assays to extensively characterize the consequences of HIF2alpha deficiency on Treg-mediated immune suppression.

As the authors rightly point out, HIF-2's role in T cells is understudied. Additionally, the literature regarding HIF-1 and Tregs to date suggests a complicated role, which the authors summarize well for the reader's benefit (i.e. HIF-1 is reported to inhibit iTreg differentiation while potentially contributing to the function of established Tregs in colitis models). As such, this manuscript is aimed to add substantially to our understanding of the impact of hypoxia and hypoxia-inducible factors on Treg function.

The authors report that while HIF-2 deficiency does not impact the development of thymic (t)Tregs in mice or the in vitro differentiation of iTregs, the authors found that the in vivo function of these cells were markedly compromised relative to that of wild type controls in across a diverse number of animal disease models. Notably, they also show that the suppressive deficiency of HIF-2-deficient Tregs results from elevated HIF-1 levels. This finding suggests a hitherto unappreciated cross-regulation between HIF-1 and HIF-2 with major consequences for Treg function.

The paper is well-written for the most part. Experimental design and execution is sound, and results are clearly presented that shed much needed light on the role played by the important hypoxia-inducible factors in the biology of Tregs and immune control. The findings reported in this manuscript will likely be viewed as a significant advancement in our understanding of HIFs and Treg biology relevant to a wide audience (including experts in cancer, autoimmunity, and transplant biology). Addressing a few (mostly minor, listed below) issues with the current version of the manuscript will strengthen the work and ensure its impact.

Major Issues:

1. The text describing the results of the iTreg differentiation experiments depicted in Figure 1c seems a little over simplified. From the representative flow plots, it appears that HIF-1 deficiency does enhance Foxp3+ cell percentages at low (17.6% vs. 26.0%) and modest TGF-beta concentrations (67.5% vs. 78.1%) relative to wild type controls. As the authors point out, hypoxia did stunt Foxp3 upregulation by wild type T cells at multiple TGF-beta concentrations. However, at the lowest TGF-beta only, hypoxia seemed to have an enhancing effect. Quantification of Foxp3 percentages in Fig. 1a,c,d and the Mean Fluorescence Intensities (MFIs) of Fig. 1b,c,d should be presented with statistics to better illustrate the reproducibility of the trends suggested by the representative plots shown. If these trends indicated by the representative plots shown are consistently seen across independent trials, and if they are statistically significant, some additional discussion/description would seem warranted. Similarly, representative plots in Fig. 2f, 3a, 4f, and 6c would benefit from quantification panels

2. It is unclear how Foxp3 was measured in Figure 2g, 6d. Was this by flow cytometry, RT-PCR on recovered cells? Additionally, what normalized parameter is shown (MFI, %+ cells)? If flow was

used to generate these plots, it would be preferable to see quantification of Foxp3+, IL-17+, and IFN γ + percentages and the MFI of each signal, with statistics instead.

3. The authors convincingly show that deletion of HIF-1 restores the Treg in vivo function of HIF-2 deficient suppressors in the colitis model, supporting the notion that compromised function stems from the ability of HIF-2 to keep HIF-1 levels in check. Corroborating findings from the airway inflammation model and/or the tumor model would substantially strengthen the paper's very intriguing finding of HIF-2:HIF-1 cross talk in Tregs.

4. Additionally, the authors should explore whether HIF-1 driven metabolic alterations are associated with the dysfunction of HIF-2-deficient Tregs, since loss of Foxp3 expression does not seem to be responsible, this would seem to be the most likely underlying mechanism. Measuring glycolysis-associated genes and factors in Tregs lacking HIF-2 expression could shed light on this possibility. Additionally, the authors should at the very least speculate on the mechanism responsible for the apparent repression of HIF-1 expression by HIF-2 in Tregs.

Minor Issues:

1. The authors findings suggest a truly unappreciated cross talk between HIF-1 and HIF-2 in Tregs that is key in determining their functional fidelity. As one of their more interesting findings, this should get a little more emphasis in the abstract.

2. Introduction, page 3, line 55: Transcriptional means of regulating the stability of Foxp3 gene expression (i.e. TSDR demethylation) should also be mentioned as well as posttranslational means. Line 56: as the full-fledged conversion of Tregs to inflammatory effectors has been a controversial subject (with some Tregs appearing very resistant to destabilization and other much less so), the authors should amend the language used here... for example stating that "Some Tregs can be converted into inflammatory effector cells following Foxp3 destabilization, ..."

3. There are a few instances of awkward phrasing/word choice through the manuscript. For instance, the last paragraph of the introduction; page 8, line 173 ("a large number of HIF2 α -/- tTregs were compromised...". Some additional language editing would benefit the paper.

4. The brief description of the adoptive transfer colitis model in the results section (bottom of page 6, page 7) should be a little more descriptive (e.g. include mention that the Tregs and the conventional CD4+ T cells injected were congenically distinct).

5. On Page 7, line 144 reference #39 is cited as an example of previous work showing a negative effect of hypoxia on iTreg development. In actually, this study by Shehade et al. showed that hypoxia could bring about the down-regulation of Foxp3 in previously differentiated iTregs. References #35 and #36, should be cited here instead, as these studies exposed developing iTregs to hypoxia and found an inhibitory effect.

6. The use of naïve T cells derived from Foxp3Cre-driven HIF1 and HIF2 conditional knockout T cells in iTreg differentiation assays like the one shown in Fig.1c,d would specifically allow the authors to draw a conclusion about the role played by each HIF post-Foxp3 induction and identify effects on stabilizing/destabilizing Foxp3 expression. Not doing so is indeed a missed opportunity.

7. The authors should avoid using Hif2 α -/- and Hif1 α -/- designations for their knockout mice. Since they use both CD4Cre- and Foxp3Cre-driven conditional knockouts in their experiments, the genotype of the specific mouse model used should be clearly stated in the figure labels, main text, and figure legends. It is currently unclear which mice were used in the experiments depicted in Figure 2 and some others. Additionally, the exact genotype of the "Wild type" controls used should also be more clearly presented (for instance, were they Hif2 α f/f Foxp3Cre- or Hif2 α wt/wt Foxp3Cre+ ? were littermate controls used?) as was done in the Figure 7 legend.

8. From the figure legend, it is unclear how many independent experiments are represented in Figures 2 and 6 (adoptive transfer colitis model). If the experiments were only been once each, as it would seem to be the case, they should be repeated at least once more.

9. The purification scheme and confirmation of the starting purity/Foxp3 levels for the tTregs used in Fig. 4 should be presented.

10. Concentrations of anti-CD3/CD28 antibodies used for T cell activation should be given in the figure legends where applicable.

11. In reference to Figure 2, when the author's state that "Penetrance of the loss of in vivo suppressive activity..." was ~60% they seem to be stating that ~60% of the recipient Rag1 mice that received HIF-2 deficient Tregs lost weight and 40% did not (as is more plainly stated in the description of Figure 6. While the transparency here is nice, It may be preferable to present this as a fraction of total mice in the group displaying a lack of Treg-mediated protection (similar to the tumor-free/total mouse fractions of Figure 7). Such fractions should be presented for all groups in the figure or figure legend depicting the results of the colitis experiments.

Reviewer #3 (Remarks to the Author):

This is a relatively interesting study investigating the role of Hypoxia inducible factor 2a in Tregs. Authors are demonstrating in 3 different inflammatory models that the lack of HIF-2a limits the immunosuppressive activity of Tregs. According to them, targeting HIF-2a in cancer could have a therapeutic potential. The collected data are not enough to support the hypothesis.

comments:

Abstract: The last 2 sentences only point out conclusions about cancer and exclude colitis and airway hypersensitivity. Although cancer is an important topic, 2 experiments do not support the conclusions.

Introduction: well-written and summarizes recent research related to the topic.

Figure 1 & 2: How could the authors explain the contradiction between in vitro and in vivo results concerning HIF-1a and HIF-2a effect on Treg suppressive activity? Why HIF-1a deficiency alters the in vitro suppressive activity but has no effect in the colitis model? Why HIF-2a alters T regs in vivo but not in vitro? This should be discussed.

Figure 3: The title of figure 3 seems to be wrong; The impaired suppressive activity of HIF-2a/- Tregs enhances allergic inflammation.

a+b) why these results are placed in the airway inflammation figure? This is confusing and not well explained.

c) Histology is not enough to assess the severity of airway inflammation. Authors should add flow cytometry experiments to analyze inflammatory cell recruitments and also BAL analysis. Also authors should assess lung respiratory functions such as lung resistance and dynamic compliance.

d) Authors are claiming that the administration of HIF-2a deficient iTreg fails to decrease the secretion of IL-4 and IL-6 (compared to WT iTreg). However, the graphs do not show statistical differences between these 2 groups. Their statistics are not relevant and supportive of the conclusion. What about IL-17 and IFN γ producing Tregs? The authors have to look at ROR γ , Tbet and Gata3 in Treg population. Any similar results in the colitis model?

e) Assessment of the relative Foxp3 expression should be explained in the result section.

Figure 4: This figure should follow figure 2 (colitis)

c+f) associated graphs are missing.

Figure 5: Western blot associated graphs are missing. Authors should assess the expression of HIF-1a in recovered Treg from the colitis and airway inflammation models.

The order of appearance of figures is confusing. Authors are switching many times in vitro to in vivo and from an inflammatory model to other models, make it hard to follow.

Figure 5e: Does a selective inhibition of HIF-1a lead to the upregulation of HIF-2a?

Discussion: The authors describing the results instead of discussing the findings and compare it to

the other studies. Discussion should be rewritten completely and is not suitable for publication.

I. Revisions made to address issues raised by Reviewer 2

Reviewer #2 (Remarks to the Author):

The manuscript by Hsu, et al., reports on the yet unexplored importance of HIF-2alpha in Regulatory T cells (Tregs). In this study the authors utilize mice with Treg-specific HIF-2 and HIF-1 alpha deficiencies in a number of in vivo disease models (colitis, airway inflammation, cancer), as well as in vitro models of Treg differentiation and function. In their in vivo studies they use flow cytometry and ex vivo functional assays to extensively characterize the consequences of HIF2alpha deficiency on Treg-mediated immune suppression.

As the authors rightly point out, HIF-2The authors report that while HIF-2 deficiency does not impact the development of thymic (t)Tregs in mice or the in vitro differentiation of iTregs, the authors found that the in vivo function of these cells were markedly compromised relative to that of wild type controls in across a diverse number of animal disease models. Notably, they also show that the suppressive deficiency of HIF-2-deficient Tregs results from elevated HIF-1 levels. This finding suggests a hitherto unappreciated cross-regulation between HIF-1 and HIF-2 with major consequences for Treg function.

The paper is well-written for the most part. Experimental design and execution is sound, and results are clearly presented that shed much needed light on the role played by the important hypoxia-inducible factors in the biology of Tregs and immune control. The findings reported in this manuscript will likely be viewed as a significant advancement in our understanding of HIFs and Treg biology relevant to a wide audience (including experts in cancer, autoimmunity, and transplant biology). Addressing a few (mostly minor, listed below) issues with the current version of the manuscript will strengthen the work and ensure its impact.

Major Issues:

1. The text describing the results of the iTreg differentiation experiments depicted in Figure 1c seems a little over simplified. From the representative flow plots, it appears that HIF-1 deficiency does enhance Foxp3+ cell percentages at low (17.6% vs. 26.0%) and modest TGF-beta concentrations (67.5% vs. 78.1%) relative to wild type controls. As the authors point out, hypoxia did stunt Foxp3 upregulation by wild type T cells at multiple TGF-beta concentrations. However, at the lowest TGF-beta only, hypoxia seemed to have an enhancing effect. Quantification of Foxp3 percentages in Fig. 1a,c,d and the Mean Fluorescence Intensities (MFIs) of Fig. 1b,c,d should be presented with statistics to better illustrate the reproducibility of the trends suggested by the representative plots shown. If these trends indicated by the representative plots shown are consistently seen across independent trials, and if they are statistically significant, some additional discussion/description would seem warranted. Similarly, representative plots in Fig. 2f, 3a, 4f, and 6c would benefit from quantification panels

1.1. Reviewer 2 has rightly pointed out the enhancing effect of hypoxia on Foxp3 expression at the low dose of TGF-beta in Fig. 1d, which we did not comment on in the original manuscript. We have repeated the experiments and followed the reviewer's suggestion to quantify Foxp3 expression in terms of both percentage Foxp3⁺ cells and mean fluorescence intensity (MFI). Data quantification revealed that there is a significant increase in the proportion of Foxp3-expressing cells among WT Tregs developed under hypoxic conditions and with 0.5 ng/ml TGF-beta treatment. The respective data and statistics are now presented in the right panel of revised Fig. 1c and Fig. 1d. The effect of hypoxia in terms of increasing WT iTreg differentiation at low-dose TGF-beta treatment is now described on page 7 of our Results, and it is extensively discussed on page 16 of the Discussion of our revised manuscript.

1.2. We have also quantitated the results for Fig. 1a and 1b. Data on the proportion of

Foxp3⁺ cells and MFI (with respective statistics) are now presented in the right panels of revised Fig. 1a and Fig. 1b.

1.3. Quantitation plots with respective statistics have also been added to revised Figs. 2e, 2f, 3f (original Fig. 4f), 6c and Supplementary Fig. 5a (original Fig. 3a),

2. It is unclear how Foxp3 was measured in Figure 2g, 6d. Was this by flow cytometry, RT-PCR on recovered cells? Additionally, what normalized parameter is shown (MFI, %+ cells)? If flow was used to generate these plots, it would be preferable to see quantification of Foxp3⁺, IL-17⁺, and IFN γ ⁺ percentages and the MFI of each signal, with statistics instead.

2. We acknowledge that our original presentations of Foxp3 measurements in Figs. 2g, 6d were confusing. In both cases, data was generated from flow cytometry. We have now revised Figs. 2g and 6d, as well as our descriptions on pages 9 and 13 (Results), to be clearer. Quantitation of proportions of Foxp3⁺, IL-17⁺ and IFN γ ⁺ cells are now presented in the revised Figs. 2e and 6c.

3. The authors convincingly show that deletion of HIF-1 restores the Treg in vivo function of HIF-2 deficient suppressors in the colitis model, supporting the notion that compromised function stems from the ability of HIF-2 to keep HIF-1 levels in check. Corroborating findings from the airway inflammation model and/or the tumor model would substantially strengthen the paper. Additionally, the authors should explore whether HIF-1 driven metabolic alterations are associated with the dysfunction of HIF-2-deficient Tregs, since loss of Foxp3 expression does not seem to be responsible, this would seem to be the most likely underlying mechanism. Measuring glycolysis-associated genes and factors in Tregs lacking HIF-2 expression could shed light on this possibility. Additionally, the authors should at the very least speculate on the mechanism responsible for the apparent repression of HIF-1 expression by HIF-2 in Tregs.

3.1. We have adopted the suggestion to determine the effect of additional deletion of HIF-1 α on HIF-2 α KO mice using an airway inflammation model. By measuring ovalbumin-induced airway resistance in primed mice, we show that allergic airway inflammation, which is resistant to the transfer of HIF-2 α -deficient iTregs, was alleviated by administration of HIF-1 α -HIF-2 α double-knockout iTregs. These results further strengthen the proposed notion that elevated HIF-1 α inhibits the suppressive activity of HIF-2 α -deficient Tregs. The new data are presented as new Figs. 6e, 6f and 6g in the revised manuscript, and they are described on page 13.

3.2. We have adopted the suggestion to analyze the expression of a panel of metabolic HIF-1 α -targeted genes. We found that *Glut1* was increased and *Pdk1* was marginally elevated in HIF-2 α -deficient Tregs, but expression of the majority of the HIF-1 α -targeted metabolic genes we assessed was not significantly increased. We speculate that the elevation of HIF-1 α in HIF-2 α -knockout Tregs was sufficiently modest that it did not result in an apparent increase of glycolytic enzymes. The results of quantitative PCR on metabolic genes are presented in new Fig. 5k and Supplementary Fig. 8, they are described on page 12 (Results), and further discussed on pages 18 and 19 (Discussion) of the revised manuscript.

3.3. We have conducted an additional experiment and included speculation on the mechanism responsible for repression of HIF-1 α by HIF-2 α in Tregs. The new data show that HIF-2 α knockout increased HIF-1 α expression in tTregs (Fig. 5e), that deficiency of HIF-2 α increased the stability of HIF-1 α protein (Fig. 5g), and that treatment of WT tTregs with the

HIF-2 α inhibitor PT2385 also led to greater production of IL-17, as observed for HIF-2 α knockout tTregs (Fig. 5j). These results are now described on page 12 of the revised manuscript. We now discuss how HIF-2 α inhibits HIF-1 α expression on page 19.

Minor Issues:

1. The authors findings suggest a truly unappreciated cross talk between HIF-1 and HIF-2 in Tregs that is key in determining their functional fidelity. As one of their more interesting findings, this should get a little more emphasis in the abstract.

1. Thank you for this suggestion. We now emphasize that HIF-2 α inhibits HIF-1 α in the Abstract of our revised manuscript.

2. Introduction, page 3, line 55: Transcriptional means of regulating the stability of Foxp3 gene expression (i.e. TSDR demethylation) should also be mentioned as well as posttranslational means. Line 56: as the full-fledged conversion of Tregs to inflammatory effectors has been a controversial subject (with some Tregs appearing very resistant to destabilization and other much less so), the authors should amend the language used here x2026; for example stating that x201C; Some Tregs can be converted into inflammatory effector cells following Foxp3 destabilization, x2026;x201D;

2. We have now revised our Introduction. We now incorporate TSDR demethylation in the part where we describe transcriptional regulation of Foxp3 expression and stability. We have also modified the language relating to conversion of Tregs into inflammatory effector cells. Both these revisions are made on page 3 (Introduction) of our revised manuscript.

3. There are a few instances of awkward phrasing/word choice through the manuscript. For instance, the last paragraph of the introduction; page 8, line 173 (x201C; a large number of HIF2alpha-/- tTregs were compromisedx2026;x201D;. Some additional language editing would benefit the paper.

3. We have now had the manuscript re-edited by a professional English editor.

4. The brief description of the adoptive transfer colitis model in the results section (bottom of page 6, page 7) should be a little more descriptive (e.g. include mention that the Tregs and the conventional CD4+ T cells injected were congenically distinct).

4. Acknowledged. We now more fully describe our adoptive transfer colitis experiment (page 8; Results).

5. On Page 7, line 144 reference #39 is cited as an example of previous work showing a negative effect of hypoxia on iTreg development. In actually, this study by Shehade et al. showed that hypoxia could bring about the down-regulation of Foxp3 in previously differentiated iTregs. References #35 and #36, should be cited here instead, as these studies exposed developing iTregs to hypoxia and found an inhibitory effect.

5. Apologies. We have now corrected this error. The suggested references (ref. #41 and #42) are now cited in the revised manuscript (Page 7).

6. The use of naive T cells derived from Foxp3Cre-driven HIF1 and HIF2 conditional knockout T cells in iTreg differentiation assays like the one shown in Fig.1c,d would specifically allow the authors to draw a conclusion about the role played by each HIF post-Foxp3 induction and identify effects on stabilizing/destabilizing Foxp3 expression. Not doing so is indeed a missed opportunity.

6. Agreed. We have now determined iTreg differentiation from Foxp3^{Cre}-driven HIF-1 α and HIF-2 α conditional knockout T cells under normoxic and hypoxic conditions. We found that hypoxia enhanced the proportion of Foxp3⁺ cells and Foxp3 MFI in Foxp3^{Cre}Hif1 α ^{ff} iTregs, relative to Foxp3^{Cre} and Foxp3^{Cre}Hif2 α ^{ff} iTregs. Therefore, an inhibitory effect of HIF-1 α could be detected post-Foxp3 induction, suggesting that HIF-1 α impacts the stability of Foxp3. These results are now presented in new Supplementary Fig. 6, they are described in the Results (on page 7), and they are discussed on pages 15 and 16 (Discussion) of the revised manuscript.

7. The authors should avoid using Hif2alpha^{-/-} and Hif1alpha^{-/-} designations for their knockout mice. Since they use both CD4Cre- and Foxp3Cre-driven conditional knockouts in their experiments, the genotype of the specific mouse model used should be clearly stated in the figure labels, main text, and figure legends. It is currently unclear which mice were used in the experiments depicted in Figure 2 and some others. Additionally, the exact genotype of the x201C;Wild type x201D controls used should also be more clearly presented (for instance, were they Hif2alpha f/f Foxp3Cre- or Hif2alpha wt/wt Foxp3Cre+ ? were littermate controls used?) as was done in the Figure 7 legend.

7. Apologies for the confusion. We now employ specific terms to indicate cell origin, such as CD4^{Cre}, Cd4^{Cre}Hif1 α ^{ff} or Cd4^{Cre}Hif2 α ^{f/f}, throughout the revised manuscript. The term WT in this instance indicates Foxp3^{Cre} littermate controls of Foxp3^{Cre}Hif2 α ^{ff} mice, which we now specify in the legends of Fig. 7 and Supplementary Fig. 10.

8. From the figure legend, it is unclear how many independent experiments are represented in Figures 2 and 6 (adoptive transfer colitis model). If the experiments were only been once each, as it would seem to be the case, they should be repeated at least once more.

8. Due to the limited numbers of mice of similar age in each experiment for colitis inhibition, the results from two to three independent repeats were combined and were presented in the respective figure. Data from each individual experiment displayed a pattern comparable to that of the combined dataset. We have included this information in the legends of Figs. 2 and 6. We have also partially repeated the experiment (see Fig. 5a) to verify the reproducibility of our results.

9. The purification scheme and confirmation of the starting purity/Foxp3 levels for the tTregs used in Fig. 4 should be presented.

9. Agreed. The purification scheme and purity of tTregs in original Fig. 4 (new Fig. 3) are now presented in Supplementary Fig. 11 of the revised manuscript.

10. Concentrations of anti-CD3/CD28 antibodies used for T cell activation should be given in the figure legends where applicable.

10. Agreed. Concentrations of anti-CD3/CD28 antibodies used in each experiment are now stated in the figure legends (Figs. 1, 3, and 5, Supplementary Fig. 6).

11. In reference to Figure 2, when the authors state that Penetrance of the loss of in vivo suppressive activity was ~60% they seem to be stating that ~60% of the recipient Rag1 mice that received HIF-2 deficient Tregs lost weight and 40% did not (as is more plainly stated in the description of Figure 6. While the transparency here is nice, It may be preferable to present this as a fraction of total mice in the group displaying a lack of Treg-mediated protection (similar to the tumor-free/total mouse fractions of Figure 7). Such fractions should be presented for all groups in the figure or figure legend depicting the results of the colitis experiments.

11. Acknowledged. We have removed the description from the Results section, and now present details of fractions of non-responsive mice in the figure legends.

Revisions made to address issues raised by Reviewer 3

This is a relatively interesting study investigating the role of Hypoxia inducible factor 2a in Tregs. Authors are demonstrating in 3 different inflammatory models that the lack of HIF-2a limits the immunosuppressive activity of Tregs. According to them, targeting HIF-2a in cancer could have a therapeutic potential. The collected data are not enough to support the hypothesis.

comments:

Abstract: The last 2 sentences only point out conclusions about cancer and exclude colitis and airway hypersensitivity. Although cancer is an important topic, 2 experiments do not support the conclusions.

Introduction: well-written and summarizes recent research related to the topic.

1. We appreciate the supportive comments of Reviewer 3 regarding our Introduction. We agree with the reviewer that we should not draw firm conclusions from our two cancer experiments (Fig. 7), so we have made the requisite modifications to our Abstract.

Figure 1 & 2: How could the authors explain the contradiction between *in vitro* and *in vivo* results concerning HIF-1 α and HIF-2 α effect on Treg suppressive activity? Why HIF-1 α deficiency alters the *in vitro* suppressive activity but has no effect in the colitis model? Why HIF-2 α alters T regs *in vivo* but not *in vitro*? This should be discussed.

2.1. We have now included a more thorough discussion of the difference between *in vitro* and *in vivo* suppression for HIF-1 α -deficient and HIF-2 α -knockout Tregs (pages 17 and 18).

2.2. Recent studies have revealed that Treg-specific knockout of several genes (such as CD28, Ubc13, Helios or Ezh2) does not affect the *in vitro* suppressive activity of Tregs, but impairs the *in vivo* inhibitory activity of Tregs (Ref #52, #58-61). HIF-2 α -deficient Tregs can be included in this group. Based on what we know about Tregs from this group, we now speculate on the possible mechanism involved in the suppressive activity of HIF-2 α -deficient Tregs (pages 17 and 18).

2.3. For HIF-1 α -deficient Tregs, the ability to suppress effector T cell activation *in vitro* but not *in vivo* is less frequently reported. We speculate that it may be associated with HIF-1 α -dependent Foxp3 expression *in vitro*, or could reflect a difference between the *in vitro* suppression assay, dominated by cell proximity, and the *in vitro* inhibitory analysis, which is more determined by specific cognation. We now discuss this possibility on page 17.

Figure 3: The title of figure 3 seems to be wrong; The impaired suppressive activity of HIF-2 α -/- Tregs enhances allergic inflammation.

a+b) why these results are placed in the airway inflammation figure? This is confusing and not well explained.

c) Histology is not enough to assess the severity of airway inflammation. Authors should add flow cytometry experiments to analyze inflammatory cell recruitments and also BAL analysis. Also authors should assess lung respiratory functions such as lung resistance and dynamic compliance.

d) Authors are claiming that the administration of HIF-2 α deficient iTreg fails to decrease the secretion of IL-4 and IL-6 (compared to WT iTreg). However, the graphs do not show statistical differences between these 2 groups. Their statistics are not relevant and supportive of the

conclusion. What about IL-17 and IFN γ producing Tregs? The authors have to look at ROR γ T, Tbet and Gata3 in Treg population. Any similar results in the colitis model?
e) Assessment of the relative Foxp3 expression should be explained in the result section.

3.1. Apologies, we have now corrected the title of Fig. 4 (original Fig. 3) to “Inability of HIF-2 α -deficient iTregs to inhibit allergic inflammation”.

3.2. Acknowledged. We have followed the suggestion and moved original Figs. 3a and 3b to Supplementary Fig. 5, and the data are now described with our *in vitro* characterization of tTregs (depicted in Fig. 1) on pages 7 and 8 of our Results.

3.3. Acknowledged. We have now conducted a more rigorous analysis of airway inflammation, including lung resistance and an analysis of leukocyte infiltration in BALF. These results are presented in new Figs. 4b, 4c and Fig. 6e, and are described on pages 10, 11 and 13 of the Results.

3.4. We also attempted to determine the effect of iTregs delivered after secondary ovalbumin sensitization. However, WT iTregs (and *Hif1a/Hif2a* double-knockout iTregs) administered in that way failed to inhibit leukocyte infiltration and cytokine expression in BALF, despite attenuating lung resistance. This result is now presented in Supplementary Fig. 9, and it is described on page 13 of the Results.

3.5. Acknowledged. We have now added statistical analysis to support the difference in suppressive effects between WT and HIF-2 α -deficient iTregs (Fig. 4d). IFN- γ was measured in BALF and the results are included in Fig. 4d. Levels of IL-17 were too low to be detected in our experimental setup.

3.6. Likely due to the low number of Tregs recovered from our airway inflammation or colitis models, we failed to detect ROR γ T, GATA3 and T-bet in Tregs. (Foxp3, IL-17, IFN- γ and HIF-1 α in recovered Tregs was detected by fluorochrome-conjugated primary antibodies). A low level of ROR γ T was detected in tTregs re-programmed *in vitro*, but was not increased in HIF-2 α -deficient tTregs (Supplementary Fig. 7).

3.7. Apologies, for the confusion caused by the original Fig. 3e (now Fig. 4g). Foxp3 level was measured by flowcytometry, is indicated in the revised Fig. 4g and described on page 11 (Results).

Figure 4: This figure should follow figure 2 (colitis) c+f) associated graphs are missing.

4. Apologies for the confusion. We have adopted the suggestion and moved original Fig. 4 to new Fig. 3. Quantitation and statistics are included in the right-most panels of Figs. 3c and 3f.

Figure 5: Western blot associated graphs are missing. Authors should assess the expression of HIF-1 α in recovered Treg from the colitis and airway inflammation models.

The order of appearance of figures is confusing. Authors are switching many times *in vitro* to *in vivo*

and from an inflammatory model to other models, make it hard to follow.

Figure 5e: Does a selective inhibition of HIF-1 α lead to the upregulation of HIF-2 α ?

5.1. Agreed. We have now revised Fig. 5 according to the reviewer's suggestion. We determined HIF-1 α levels in Tregs recovered from our colitis model, which showed that HIF-1 α was higher in the recovered *Hif2 α ^{-/-}* tTregs. These data are presented in new Fig. 5a, and they are described on page 11. However, the number of CD45.2⁺ tTregs recovered from *Rag1^{-/-}* mice was very low, only in the hundreds per mouse, so we did not have sufficient cells to perform a Western blot. Likely due to the long interval (25 days) between iTreg administration and lung dissection, recovery of iTregs from our airway inflammation model was also too low for Western blot analysis.

5.2. Apologies for the confusion regarding original Fig. 5. We have attempted to present our results in a more logical order in the revised manuscript.

5.3. We have now included results showing that *Hif2 α* transcript was also up-regulated in *Hif1 α* -knockout Tregs, suggesting mutual regulation between HIF-1 α and HIF-2 α . These results are presented in new Fig. 5l, they are described on page 12 (Results), and are discussed on page 19 (Discussion). Regrettably, we are not aware of an inhibitor that can inhibit HIF-1 α as specifically as how PT2385 inhibits HIF-2 α .

Discussion: The authors describing the results instead of discussing the findings and compare it to the other studies. Discussion should be rewritten completely and is not suitable for publication.

6. We have now extensively revised the Discussion, with a particular emphasis on comparing our findings to other studies.

REVIEWERS' COMMENTS:

Reviewer #3 (Remarks to the Author):

The authors sufficiently addressed the concerns raised in my previous review.

Reviewer #4 (Remarks to the Author):

The authors have successfully addressed the previous comments.

Revisions made to address issues raised by Reviewers

REVIEWERS' COMMENTS:

[REDACTED]

Reviewer #3 (Remarks to the Author):

The authors sufficiently addressed the concerns raised in my previous review.

Reviewer #4 (Remarks to the Author):

The authors have successfully addressed the previous comments.

We thank reviewers for their efforts and positive remarks.